

# Investigation of meteorological conditions and BrO during Ozone Depletion Events in Ny-Ålesund between 2010 and 2021

Bianca Zilker[1], Andreas Richter[1], Anne-Marlene Blechschmidt[1], Peter von der Gathen[2],
Ilias Bougoudis[3], Sora Seo[4], Tim Bösch[1], and John Philip Burrows[1]

[1]Institute of Environmental Physics, University of Bremen, Bremen, Germany
[2]Alfred Wegener Institute, Helmholtz Centre for Polar and Marine Research, Potsdam, Germany
[3]LuftBlick OG, Innsbruck, Austria
[4]Deutsches Zentrum für Luft- und Raumfahrt, Institut für Physik der Atmosphäre, Oberpfaffenhofen, Germany

**Correspondence:** B. Zilker (bianca.zilker@iup.physik.uni-bremen.de)

**Abstract.** During polar spring, Ozone Depletion Events (ODEs) are often observed in combination with Bromine Explosion Events (BEEs) in Ny-Ålesund. In this study, two long term ozone data sets (2010-2021) from ozone sonde launches and in-situ ozone measurements have been evaluated between March and May of each year, to study ODEs in Ny-Ålesund. Ozone concentrations below 15 ppb were marked as ODE. We applied a composite analysis to evaluate tropospheric BrO retrieved

from satellite data and the prevailing meteorological conditions during these events. During ODEs, both data sets show a blocking situation with a low pressure anomaly over the Barents Sea and anomalously high pressure in the Icelandic low area, leading to transport of cold polar air from the north to Ny-Ålesund with negative temperature and positive BrO anomalies found around Svalbard. Also higher wind speed and a higher, less stable boundary layer are noticed, supporting the assumption that ODEs often occur in combination with polar cyclones. Applying a 20 ppb ozone threshold value to tag ODEs resulted in only

a slight attenuation of the BrO and meteorological anomalies compared to the 15 ppb threshold. Monthly analysis showed that BrO and meteorological anomalies are weakening from March to May. Therefore, ODEs associated with low pressure systems, high wind speeds and blowing snow more likely occur in early spring, while ODEs associated with low wind speed and stable boundary layer meteorological conditions seem to occur more often in late spring. In an annual evaluation, similar prevailing meteorological conditions were found for several years as well as in the overall result of the composite analysis.

However, some years show different meteorological patterns deviating from the results of the mean analysis. Finally, an ODE case study from the beginning of April 2020 in Ny-Ålesund is presented, where ozone was depleted for two consecutive days in combination with increased BrO values. The meteorological conditions are representative of the results of the composite analysis. A low pressure system arrived from the north-east to Svalbard resulting in high wind speeds with blowing snow and transport of cold polar air from the north.

# 1 Introduction

During polar spring, large losses of ozone in the lower troposphere are observed regularly. These 'Ozone Depletion Events' (ODEs) occur often in combination with enhanced bromine monoxide (BrO) values. In the 1980s, Barrie et al. (1988) first





reported a link between the ozone loss and the appearance of reactive bromine, further known as 'Bromine Explosion Events'
(BEEs) (Barrie and Platt, 1997), and concluded that ozone destruction takes place during sunlight in autocatalytic chain re-
actions involving $BrO_x$ (Br, BrO) radicals. Usually, these events last between a couple of hours and a few days, where ozone
can be depleted below detection limit of the instruments (e.g., Langendörfer et al., 1999; Jacobi et al., 2006). The current un-
derstanding of this ozone depletion is that in a heterogeneous, autocatalytic, chemical chain reaction cycle, molecular bromine
($Br_2$) is released from the cryosphere (e.g. sea ice, brine, snow) into the troposphere (e.g., Simpson et al., 2007). In the presence
of sunlight, $Br_2$ is photolyzed creating bromine (Br) radicals (Reaction R1) that react with ozone to yield bromine oxide, BrO
(Reaction R2). The resulting BrO can be observed by satellite and ground-based measurements using the Differential Optical
Absorption Spectroscopy (DOAS) technique (Platt and Stutz, 2008). After the oxidation, BrO reacts with $HO_2$ and forms HOBr
(Reaction R3) which is scavenged and enters the cryosphere again, converted from the gas phase into a liquid or solid phase,
respectively (Reaction R4). In the liquid phase, HOBr reacts with halogens to form $Br_2$ (Reaction R5), which is released to the
atmosphere, and the cycle starts over again. As for each Br atom from the gas phase, two Br atoms are released from the liquid
phase, the mechanism is autocatalytic and rapidly accelerating.

$$Br_2 + h\nu \rightarrow 2Br \tag{R1}$$

$$2Br + 2O_3 \rightarrow 2BrO + 2O_2 \tag{R2}$$

$$2BrO + 2HO_2 \rightarrow 2HOBr + 2O_2 \tag{R3}$$

$$2HOBr(g) \leftrightarrow 2HOBr(s,l) \tag{R4}$$

$$2HOBr(s,l) + 2Br^-(s,l) + 2H^+(s,l) \rightarrow 2H_2O + 2Br_2 \tag{R5}$$


$$Net: 2O_3 + 2HO_2 + 2Br^- + H^+ \rightarrow 2H_2O + 4O_2 + 2Br_2 \tag{R6}$$

There is also a potential catalytic depletion cycle involving bromine chloride (BrCl). However, chlorine (Cl) atoms, unlike
Br atoms, react with methane ($CH_4$) and alkyl Volatile Organic Carbon (VOC) compounds in the troposphere, reducing the Cl
atoms concentrations available for further reactions in the catalytic cycle.



There are still uncertainties about the sources of the bromide ion Br⁻ (Reaction R4) in the cryosphere and the initial release
mechanism of $Br_2$ that leads to the large BrO plumes observed in polar spring. So far, it is known that sea ice and saline sur-
faces are associated with the Br source. Since first year sea ice is saltier than multi-year sea ice, it is regarded as a more likely
contributor to bromine release (Simpson et al., 2007). Other potential sources include frost flowers (Kaleschke et al., 2004),
sea salt aerosols from blowing snow (Yang et al., 2010), sea spray from open leads (Peterson et al., 2015), as well as BrO

transported down from the stratosphere into the troposphere (Salawitch et al., 2010). Furthermore, low temperatures favour
the bromine explosion reaction and a pH below 6.5 seems to be necessary to start the activation of the autocatalytic reaction
cycle described above (Fickert et al., 1999). The potential triggering of the acid catalysed bromine explosion by carbonate
precipitation in cold brine and the temperature dependence of Reaction R5 has also been discussed in Sander et al. (2006). In
addition to bromine, iodine also appears to play an important role in ozone depletion in spring and is present all year round as

long as there is sunlight (Benavent et al., 2022).

Regarding the meteorological conditions leading to ODEs or BEEs, respectively, Jones et al. (2009) found two different situa-
tions favoring the events: 1. low wind speeds and a stable boundary layer, where bromine can accumulate and deplete ozone;
2. high wind speeds above approximately 10 m/s with blowing snow and a higher, unstable boundary layer. The first condition
was observed, for example, during a ship cruise west and north-west of Svalbard in 2003 (Jacobi et al., 2006). Here, the ozone

concentration was near or below the instrument's detection limit for more than 5 days, along with elevated BrO concentrations
and a stable shallow boundary layer with low wind speeds. Therefore they concluded that this event was local and no trans-
port of ozone-depleted air took place. The second condition often occurs in combination with polar cyclones, where bromine
can be recycled aloft on snow and aerosol surfaces (Peterson et al., 2017). The lifetime of BrO is a few hours. Nevertheless,
several studies (e.g., Begoin et al., 2010; Blechschmidt et al., 2016; Zhao et al., 2016) showed BrO plumes in combination

with polar cyclones which were transported over the Arctic region for several days, leading to the assumption that BrO must
have been recycled on blowing snow or aerosol surfaces, increasing the lifetime of the BrO plumes. Due to a good spatial
coverage, large BrO plumes are best observed from satellites. The first observations were reported in 1998 using the GOME
instrument (Wagner and Platt (1998), Richter et al. (1998), Chance (1998)). Subsequently, polar tropospheric BrO columns
were also retrieved from measurements of the SCIAMACHY instrument (Jacobi et al., 2006), OMI (Salawitch et al., 2010),

GOME-2 (Begoin et al., 2010; Theys et al., 2011) and TROPOMI (Seo et al., 2019). Recently, a long-term tropospheric BrO
data set (1996 to 2017) was established by Bougoudis et al. (2020) using observations from 4 different satellites. It shows an
increasing trend of tropospheric BrO of around 1.5% per year during polar springs for the evaluated time period. Seo et al.
(2020) investigated the relation between areas of elevated total BrO columns and meteorological parameters in the Arctic and
Antarctic sea ice regions using 10 years of GOME-2 measurements. In a statistical analysis, meteorological conditions were

compared to a 10-year mean during enhanced BrO periods and distinct spatial patterns were identified. They found an atmo-
spheric low pressure, with low temperature, high wind speed, and a low tropopause height prevailed in areas of enhanced BrO.
In this study, a similar approach is applied, but instead of examining large-scale regions with elevated BrO, we evaluate time
periods of ozone-depleted air at a specific location to identify the prevailing meteorological conditions during ODEs.

This study focuses on ODEs recorded in Ny-Ålesund, Svalbard (Norway), a small town on the shore of the bay of Kongs-





fjorden. For almost one decade, the bay has not been completely frozen in the winter months (Serrat, 2015). Mainly icebergs and patches of ice sheets occur in the otherwise ice-free bay during winter and early spring. Ny-Ålesund is the northernmost permanent civilian research station and several institutes have installed measurement facilities and instruments on-site, which gives the opportunity to examine ODEs from several aspects. The first studies of ODEs in combination with increased bromine values in the Ny-Ålesund area were published as part of the ARCTOC (Arctic Tropospheric Ozone Chemistry) campaigns in

1995 and 1996 (e.g., Tuckermann et al., 1997; Langendörfer et al., 1999). Luo et al. (2018) observed enhanced BrO values in combination with depleted ozone in Ny-Ålesund using a ground-based MAX-DOAS instrument. Sea ice, which was transported to Kongsfjorden during that time, was assumed to be the major source of $Br_2$. Chen et al. (2022) recently reported a BEE in Ny-Ålesund which was associated with a polar cyclone approaching Svalbard, leading to high wind speeds and blowing snow. Here, it was assumed that the observed BrO was transported and recycled on its way to Ny-Ålesund.

In this study, the occurrence of ODEs in Ny-Ålesund and their dependence on meteorological conditions is investigated. To this end, ozone data measured between 2010 and 2021 in Ny-Ålesund are combined with satellite and ground-based BrO observations, sea ice concentrations and meteorological parameters in a composite analysis. In addition, a case study of a severe ODE observed in Ny-Ålesund preceded by a polar cyclone in the beginning of April 2020 is presented.

The paper is structured as follows: in section 2, all data sets used and the composite analysis are introduced. In section 3, the

results of the composite analysis are described and discussed (section 3.1). This is complemented by a sensitivity study in section 3.2, which analyses the influence of different threshold values (section 3.2.1) and investigates whether the results of the composite analysis are only valid in given months (section 3.2.2) or years (section 3.2.3). In section 3.3, the observations of a case study are discussed. The paper ends with a summary and conclusion (section 4).

## 2 Data and methods

### 2.1 Ozone data

For this study, two ozone data sets have been used to analyse ODEs in Ny-Ålesund. The first data set originates from ozone sondes that have been launched from Ny-Ålesund at least once per week since 1992. The sounding frequency increases up to about two to three times per week plus additional launches during campaigns. Data for ozone sonde records are publicly available from the Network for the Detection of Atmospheric Composition Change (NDACC) at https://www.ndacc.org (last

access: 16 February 2022). Due to the vertical resolution of ozone sondes, the altitude profile of ODEs in the boundary layer and troposphere can be studied. The majority of ODEs probably occurs in the lower troposphere. Consequently, the ozone sonde data was evaluated from the surface up to 4 km. An examination of the ozone sonde data above 4 km revealed no ozone concentrations below the two applied ODE threshold values.

To overcome the irregular temporal coverage of the ozone sonde record and allow for better statistics, additional in-situ ozone

measurements from the Zeppelin observatory on top of the Zeppelin mountain (474 m) about 1 km south of Ny-Ålesund are included in our analyses. Hence, the observatory is located higher above sea level and not necessarily capture all ODEs at sea level. Ozone concentrations have been monitored there almost continuously at hourly resolution since October 1988 and are





**Table 1.** The number of occurrences of ODEs and no ODEs for each month and year from ozone sonde observations

| Year | March | | April | | May | | Total | |
|------|------|--------|------|--------|------|--------|------|--------|
| | ODE | no ODE | ODE | no ODE | ODE | no ODE | ODE | no ODE |
| 2010 | 0 | 11 | 0 | 6 | 0 | 4 | 0 | 21 |
| 2011 | 1 | 16 | 1 | 9 | 0 | 4 | 2 | 29 |
| 2012 | 0 | 6 | 0 | 4 | 1 | 4 | 1 | 14 |
| 2013 | 0 | 3 | 0 | 4 | 0 | 2 | 0 | 9 |
| 2014 | 0 | 11 | 1 | 3 | 2 | 3 | 3 | 17 |
| 2015 | 0 | 12 | 0 | 5 | 0 | 4 | 0 | 21 |
| 2016 | 1 | 10 | 1 | 6 | 0 | 4 | 2 | 20 |
| 2017 | 0 | 6 | 0 | 4 | 0 | 3 | 0 | 13 |
| 2018 | 0 | 10 | 0 | 5 | 0 | 5 | 0 | 20 |
| 2019 | 0 | 9 | 1 | 6 | 0 | 5 | 1 | 20 |
| 2020 | 2 | 13 | 3 | 11 | 0 | 4 | 5 | 28 |
| 2021 | 0 | 9 | 0 | 3 | 0 | 4 | 0 | 16 |
| **Total** | 4 | 116 | 7 | 66 | 3 | 46 | **14** | **228** |

provided by the Norwegian Air Research Institute (Platt et al., 2022). This dataset enables a much better temporal resolution of ODEs than the ozone sondes, which is helpful for the statistical analysis later, but due to the fixed location, no altitude profiles are possible.

Since ODEs occur mainly in polar spring, both data sets are evaluated between March and May for each year from 2010-2021. The data is separated into ODEs and no ODEs using a threshold value of 15 ppb. The sensitivity to the choice of the threshold value is further discussed in section 3.2. Both data sets are shown in Figure 1. All 14 ozone sondes that contain ozone values below 15 ppb at altitudes between 0 and 2 km are marked as ODE and displayed in red. The remaining ozone sondes are coloured in blue. The background level of ozone is normally around 40 ppb in the boundary layer region. In Figure 1 all hours of the selected time period of the Zeppelin ozone data are displayed. Data points below the red 15 ppb threshold are considered as ODEs, those above are not considered to be ODEs.

Overall, 242 measurements were taken by ozone sondes and 24406 hours of ozone were monitored at the Zeppelin observatory. The frequencies of ODEs and no ODEs are listed in Table 1 for the sonde and Table 2 for the Zeppelin data set. Note that Zeppelin ozone data was not available between March 18 and March 29, 2016. Interestingly, on Zeppelin mountain, most ODEs were observed in May and also the highest number of ODEs per month for one year was observed in May 2014 with 200 ODEs. For the ozone sonde data set, most ODEs were found in April. This is probably the result of the decreasing number of sonde launches from March to May, reducing the probability of detecting ODEs in May. The influence of the individual years and months on the composite analysis are discussed in more detail in section 3.2.





**Table 2.** The number of occurrences of ODE and no ODE for each month and year from hourly in-situ ozone measurements at Zeppelin observatory.

| Year | March | | April | | May | | Total | |
|---|---|---|---|---|---|---|---|---|
| | ODE | no ODE | ODE | no ODE | ODE | no ODE | ODE | no ODE |
| 2010 | 0 | 720 | 27 | 690 | 0 | 738 | 27 | 2148 |
| 2011 | 21 | 720 | 5 | 713 | 80 | 518 | 106 | 2014 |
| 2012 | 6 | 737 | 15 | 536 | 36 | 705 | 57 | 1978 |
| 2013 | 10 | 727 | 99 | 618 | 39 | 702 | 148 | 2047 |
| 2014 | 24 | 708 | 105 | 612 | 200 | 539 | 329 | 1859 |
| 2015 | 0 | 739 | 46 | 673 | 17 | 721 | 63 | 2133 |
| 2016 | 20 | 458 | 17 | 583 | 0 | 743 | 37 | 1784 |
| 2017 | 54 | 680 | 6 | 648 | 104 | 638 | 164 | 1966 |
| 2018 | 0 | 738 | 0 | 717 | 0 | 743 | 0 | 2198 |
| 2019 | 14 | 724 | 46 | 673 | 56 | 687 | 116 | 2084 |
| 2020 | 57 | 681 | 78 | 641 | 22 | 719 | 157 | 2041 |
| 2021 | 17 | 723 | 14 | 698 | 2 | 733 | 33 | 2154 |
| **Total** | 223 | 8355 | 458 | 7802 | 556 | 8249 | **1237** | **24409** |

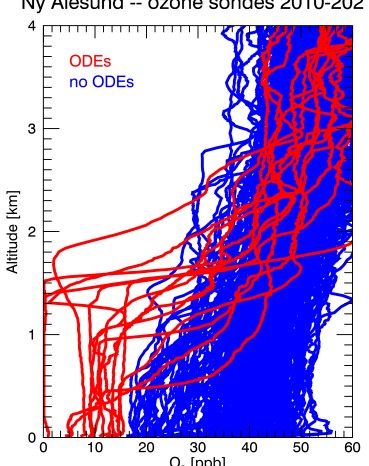

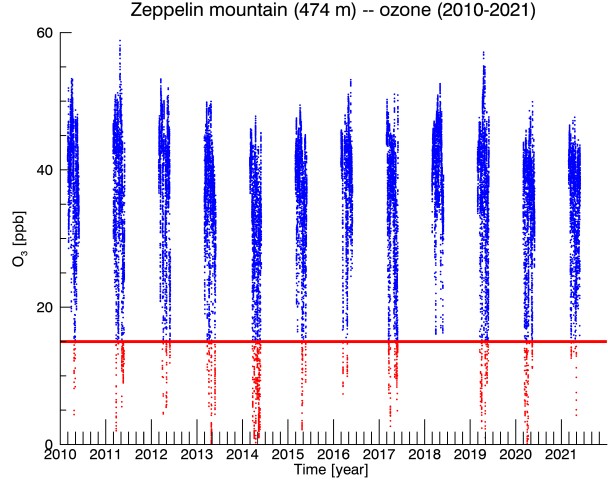

**Figure 1.** (left) All ozone launches between March and May from 2010 until 2021. Every launch with values below 15 ppb between 0-2 km is marked in red as ODE, the other launches are shown in blue. (right) All hourly ozone measurements from Zeppelin observatory for the same time period as the ozone sondes. Every hour with measurements below the red line (15 ppb) is labelled as ODE.



## 2.2 MAX-DOAS BrO profiles from Ny-Ålesund

The IUP Bremen has been operating a DOAS system on top of the roof of the observatory building of AWIPEV research
station in Ny-Ålesund since 1995 (Wittrock et al., 2000). The setup was updated to a two channel Multi AXis DOAS (MAX-
DOAS) system in 1999 (Wittrock et al., 2004). In short, the MAX-DOAS instrument consists of a telescope unit mounted on
a Pan-Tilt-Head on top of the roof of the observatory which is connected via light fibre to a spectrometer/CCD unit inside the
laboratory. Since 1999, routine measurements of scattered sun light have been performed in several azimuthal directions and
several elevation angles, enabling the analysis of trace gases in a high temporal and spatial resolution. In this study, vertical
profiles of BrO concentrations are presented which have been calculated with the inversion algorithm BOREAS (Bösch et al.,
2018) from dSCD (differential Slant Column Density) retrieved from elevation scans measured in north-westerly direction
(328° clockwise from North).

## 2.3 Satellite observations

To analyse the distribution of tropospheric BrO in the area of Svalbard during ODEs, three satellite data sets were used. The
first data set from the long-term BrO time series of Bougoudis et al. (2020) covers the time period between 2010 and 2017. In
order to fill the gap for spring 2018, GOME-2B (Global Ozone Monitoring Experiment-2B) BrO data is used. The time period
from 2019 until 2021 is covered by tropospheric BrO from TROPOMI (TROPOspheric Monitoring Instrument). All three data
sets are described in more detail below. To obtain the tropospheric SCD (Slant Column Density) of the total BrO column, a
stratospheric correction was applied based on Theys et al. (2011). To achieve a consistent BrO analysis, all three data sets were
gridded on a 0.125 ° × 0.125 ° grid. A sea ice flagging with EASE-Grid Sea Ice Age data was applied (Tschudi et al., 2019), to
analyse only sea ice covered areas.

From 2010 until 2017, BrO data from the long-term tropospheric BrO time series of Bougoudis et al. (2020) is used here. For
this data set, tropospheric BrO north of 70°N was derived for a 22-year time period (1996-2017) from four ultraviolet-visible
satellite instruments: GOME (Global Ozone Monitoring Experiment), SCIAMACHY (SCanning Imaging Absorption spec-
troMeter for Atmospheric CartograpHY), GOME-2A, and GOME-2B. The individual retrieval data sets were compared during
overlapping periods and showed good agreement (correlation of 0.82-0.98).

For spring 2018, the GOME-2B tropospheric BrO data set is used. GOME-2B is an instrument on board of the MetOp-B (Me-
teorological Operational Satellite-B) satellite and belongs to a series of three identical GOME-2 instruments on the platforms
MetOp-A, MetOp-B, and MetOp-C (Callies et al., 2000). MetOp-A with GOME-2A on board was launched in 2006, MetOp-B
with GOME-2B in 2012 and MetOp-C with GOME-2C in 2018 (Munro et al., 2016). Except for GOME-2A, the instruments
are still in operation. All three satellites were launched into a Sun-synchronous orbit with an equator-crossing time at around
9:30 LT in descending node. The GOME-2 instruments comprise identical UV, visible, and near infrared spectrometers, which
measure the upwelling radiance from the atmosphere and the solar irradiance contiguously from 232 to 790 nm. The nadir-
viewing measurement geometry covers a footprint of $80 \times 40 \text{ km}^2$ along track with a swath of 1920 km. The tropospheric
BrO VCDs (Vertical Column Densities) have been calculated according to (Bougoudis et al., 2020).



From 2019 until 2021, tropospheric BrO VCDs retrieved from the TROPOMI instrument were used. The TROPOMI (TRO-POspheric Monitoring Instrument) on board of the Copernicus S5P (Sentinel 5 Precursor) satellite was launched in October

2017 into a sun-synchronous orbit with an equator overpass at about 13:30 LT in ascending node (Veefkind et al., 2012). The push-broom nadir-viewing spectrometer with a 2600 km swath in combination with an orbit duration of about 100 min enables a daily global coverage with several overpasses over one location in the polar region. The spatial resolution of the TROPOMI pixels has been reduced from $3.5 \times 7$ km$^2$ to $3.5 \times 5.5$ km$^2$ since August 2019. TROPOMI contains 4 spectrometers, each with 2 spectral bands in the ultraviolet (270–320 nm), visible (310–500 nm), near-infrared (675–775 nm), and shortwave in-

frared (2305–2385 nm) regions. Band 3 covers the spectral range of 320-405 nm which has been used for the tropospheric BrO retrieval in this study based on the method from Seo et al. (2019). Although BrO data from TROPOMI is already available for 2018, this data is not used here due to several data gaps.

## 2.4   Meteorological and sea ice data

To investigate the meteorological parameters during ODEs, hourly ECMWF ERA5 reanalysis products have been used (Hers-

bach et al., 2020). Based on earlier studies, the following parameters have been selected: mean sea level pressure (MSLP), planetary boundary layer height (PBLH), temperature at 2 m altitude, and wind speed at 10 m altitude. The data is provided on a latitude-longitude grid of 0.25 $°\times$ 0.25 $°$.

Additionally, to analyse the sea ice concentration (SIC) around Svalbard as the potential sources region for BEE, daily AMSR (Advanced Microwave Scanning Radiometer) SIC observations on a $25 \times 25$ km$^2$ grid have been used (data accessed via

https://doi.org/10.24381/cds.3cd8b812, latest access: 22 July 2022).

## 2.5   WRF

In order to study the meteorological conditions for the ODE case study in the beginning of April 2020 and to provide wind fields for the subsequently conducted FLEXPART-WRF runs, the regional Weather Research and Forecasting (WRF) model from the National Center for Atmospheric Research (NCAR) was used (Skamarock et al., 2019). WRF is a mesoscale numerical

weather prediction and atmospheric simulation system and for this case study, version 4.2 was used.

To capture the full development of the ODE during the case study, the model run starts on 31 March 2020 at 12:00 UTC and ends on 3 April 2020 at 12:00 UTC. The model run was initialised using NCEP (National Centers for Environmental Prediction) FNL (Final) Operational Global Analysis data with a 6 hour time step and 1 $°$ spatial resolution (data accessed via https://doi.org/10.5065/D6M043C6, last access: 11 April 2022). To have a reasonable pixel size compared to the TROPOMI

resolution, two domains were used in a two-way nested run, i.e. the values of the coarse domain are overwritten by the values of the higher resolution domain at the corresponding areas. The first domain has a size of $6400 \times 6400$ km$^2$ with $20 \times 20$ km$^2$ resolution centred over the Pole covering the whole Arctic region. The second domain is located within the first one and contains the region of interest in a $3924 \times 3924$ km$^2$ domain with a $4 \times 4$ km$^2$ resolution centred north of Svalbard. The WRF output is given in 30 min time steps. The planetary boundary layer scheme from Mellor-Yamada-Janjic was applied in

the model set-up (Janjić, 1994).





## 2.6 FLEXPART-WRF

The Lagrangian trajectory model FLEXPART-WRF (Brioude et al., 2013) (https://www.flexpart.eu/), a version of the model FLEXPART (Stohl et al., 2005) driven by WRF meteorological output data, is applied to track the route of the air masses before their arrival in Ny-Ålesund by using particles released in the model. Therefore, the model runs backward in time using
the wind fields of two WRF model runs. The first run is very similar to the one described above (see section 2.5), but with a different start and end time (here: 30 March 2020, 12:00 UTC until 2 April 2020, 12:00 UTC). For the second one, the same WRF run as described above is used. To define the release altitude and time of the two FLEXPART-WRF runs, the two ozone sonde measurements of April 2nd and April 3rd are used. In total, 200000 particles are released in one grid box (about $50 \times 50$ m) containing the ozone sonde release station at the altitude region, where ozone values below 15 ppb were measured. The
FLEXPART-WRF output is written on the same spatial grid as the WRF wind fields and at a 30 min time step.

## 2.7 Method: composite analysis

To investigate the anomalies of 1. meteorological conditions, 2. BrO, and 3. SIC during ODEs, a composite analysis was conducted. For this, the two ozone data sets were evaluated individually to obtain the time points of ODEs and no ODEs for each data set. In a second step, these time points were used to calculate the anomalies of the above mentioned parameters for
ODE and no ODEs. To separate the ozone data into ODEs and no ODEs, a threshold of 15 ppb was defined and every data point below the threshold was labelled as ODE. This leads to 14 ODE and 228 no ODE days in the ozone sonde data set (see Table 1) and 1238 ODE and 24407 no ODE hours in the Zeppelin data set (see Table 2). The time points from the ozone sonde or Zeppelin data, when an ODE was measured, were used to select the $Y$ value closest in time ($Y$ represents the meteorological parameter, BrO or SIC, respectively). To obtain the mean parameter value over all ODEs $\overline{Y}_{ODE}$, all selected $Y$ were averaged.
The same procedure was applied to calculate the no ODE parameter mean $\overline{Y}_{noODE}$, except that the time points, where no ODE was measured, were used. To obtain the anomalies between ODEs and no ODEs, the averaged values of the no ODE data points ($\overline{Y}_{noODE}$) were subtracted from the averaged values of the ODE data points ($\overline{Y}_{ODE}$):

$$Y_{ODEanom} = \overline{Y}_{ODE} - \overline{Y}_{noODE}. \tag{1}$$

To investigate the development of meteorological conditions and BrO, the anomalies $Y^*_{ODEanom}$ one and two days before
and after an ODE were calculated for every individual day. To calculate the new time points, 24 hours or 48 hours were added to or subtracted from the measured ODE time point. The obtained time points were used again to select the values of the analysed parameters $Y^*$ (metrological conditions, BrO) closest in time. To obtain $\overline{Y}^*_{ODE}$, all selected $Y^*$ were averaged. The anomalies $Y^*_{ODEanom}$ were calculated by subtracting $\overline{Y}^*_{ODE}$ from $\overline{Y}_{noODE}$:

$$Y^*_{ODEanom} = \overline{Y}^*_{ODE} - \overline{Y}_{noODE}. \tag{2}$$





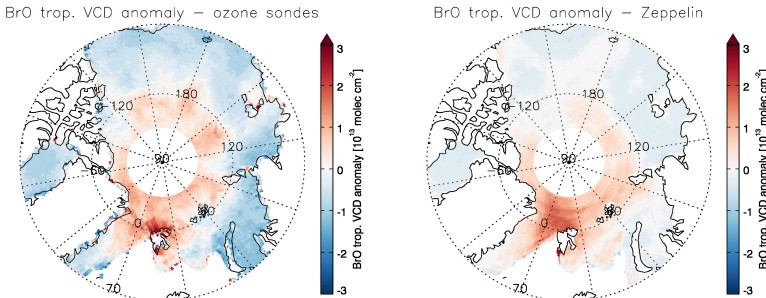

**Figure 2.** Tropospheric BrO VCD anomalies for ODE and no ODE data points using a ozone threshold value of 15 ppb, based on the (left) ozone sonde data set and (right) Zeppelin data set.

## 3 Results and discussion

### 3.1 BrO and meteorological conditions during ODEs

In this section, the anomalies of BrO, meteorological conditions, and SIC during ODEs, calculated using equation 1, are discussed. The BrO anomalies in Figure 2 show enhanced tropospheric BrO VCDs during ODEs in Ny-Ålesund for both data sets. Not only the area north of Svalbard shows a positive BrO VCD anomaly of about $2.0 \times 10^{13}$ molec cm$^{-2}$, but also in the Arctic region overall, slightly increased BrO values are noticeable. This shows that during ODEs in Ny-Alesund, BrO levels are enhanced, both locally and in the larger Arctic region.

As already mentioned in section 1, there are two meterological conditions stated by Jones et al. (2009), favoring the release of Br$_2$ from the cryosphere and thus BEEs. On the one hand, low wind speeds and a stable boundary layer, on the other hand, high wind speeds with blowing snow and a higher, unstable boundary layer. Using the composite analysis, the systematic anomalies in meteorological parameters during ODEs in Ny-Alesund can be investigated. The results are summarised in Figure 3 for both ozone data sets. The first row in Figure 3 shows that during ODEs, a low pressure anomaly is located over the Barents Sea, while pressure in the the Icelandic low area is anomalously high. Normally the Icelandic low would propagate in a north-easterly direction towards the Barents Sea, transporting warmer air from the south to Svalbard. But due to the lower pressure over the Barents Sea, the north-east propagation is blocked and cold air from the north is now transported to Svalbard. Therefore, this blocking situation due to the anomalously low pressure in the Barents Sea is associated with cold air outbreaks near Svalbard and leads to transport of cold polar air from the north to Ny-Ålesund. The transport route can also be seen in the increased wind speed (second row in Figure 3) north of Greenland and southwards into the Fram Strait (area between Greenland and Svalbard). The maximum of the wind speed anomaly is more enhanced in the ozone sonde data set and shows a maximum of about 4 ms$^{-1}$ west of Svalbard as well as north of Canada. The maximum in the Zeppelin data set is located at the northeast coast of Greenland with increased values west of Svalbard. Seo et al. (2020) also showed increased surface wind speed during enhanced BrO periods over the eastern coast of Greenland and prevailing wind directions from the north and west. These findings indicate, that Br is likely recycled on aerosol or blowing snow on its way to Ny-Ålesund, and therefore already ozone





poor air is transported to the measurement station. A lifting at the front of the low pressure system leading to an increase in the PBLH can be seen in the third row. Interestingly, the temperature anomalies show low values during ODEs only around
the area of Svalbard, whereas in the Zeppelin data, the whole northern part of the Arctic is even slightly warmer. Our findings align with the results in Seo et al. (2020), who found slightly positive temperature anomalies in the central Arctic region and lower temperatures over the Svalbard area during enhanced BrO events. The lower temperatures over the Svalbard area favour the BEE reaction cycle mechanism (Sander et al., 2006) and thus contribute to the depletion of ozone. It should be taken into account, that the overall temperature increases from March until May and therefore seasonal effects might also play a role here,
which is further discussed in section 3.2.

In addition, the behaviour of sea ice concentration (SIC) during ODEs was investigated. The cryosphere and especially salty snow or sea ice, such as freshly formed sea ice, are assumed to be an important source of $Br_2$. Figure 4 shows an increase of the SIC on ODE days for both data sets, which would be consistent with $Br_2$ sources in this area. However, seasonal effects must be taken into account, since the anomalies are also visible shortly before and after an ODE and can therefore not be directly
attributed to an ODE. The figure also includes the sea ice edge (SIC below 15 %) for no ODEs (solid line) and ODEs (dashed line). The sea ice edge (SIE) based on the ozone sonde data (left) is further south, closer to Ny-Ålesund than in the Zeppelin data (right). Possible reasons for the shift of the SIE could be: 1. The formation of fresh sea ice due to cold polar air from the north; 2. The strong winds from the north pushing the SIE further south. A combination of both is also possible.

In general, the anomaly patterns are similar between the two data sets, but less pronounced in the Zeppelin data set. This may
be due to the larger number of measurements and better coverage in April and May, which also might include more ODEs resulting from low wind speeds and a low, stable boundary layer.

To investigate the development of the meteorological conditions and BrO before and after an ODE, the anomalies were additionally examined one and two days before and after an ODE based on equation 2. Due to long computing times for the
Zeppelin data set, only the ozone sonde data is used in this analysis. The resulting figures are shown in the appendix. For BrO (Figure A1), an increase towards the day of the ODEs can be observed as well as a decrease during the days thereafter. The highest BrO anomaly over the five day period is observed mainly north of the Svalbard coast. The MSLP in Figure A2 shows a lower pressure over and east of Svalbard on the two days before the ODEs. The lower pressure migrates to the southwest throughout the examination period. On the days of the ODEs, the west coast of Svalbard lies in between the higher and lower
pressure areas. In the days after the ODE, the lower pressure moves further southeast, whereas higher pressure from the west migrates over Svalbard. Figure A3 shows an increase of wind speed towards the days of the ODEs, with a maximum of the wind speed anomaly in the area around Ny-Ålesund. On the ODE days, a maximum of the wind speed anomaly in the area around Ny-Ålesund is observed. On the days after the ODE, only minor wind speed anomalies are visible. The same behaviour is observed for the PBLH (Figure A4). A maximum anomaly occurs on the ODE days with an increasing trend towards these
days and a decrease afterwards. The temperature does not show a clear trend before and after the ODE days. However, around the days of ODEs, there are generally lower temperatures in the area around Svalbard.

This leads to the assumption that ODEs in Ny-Ålesund are often induced by polar cyclones migrating from the north-east



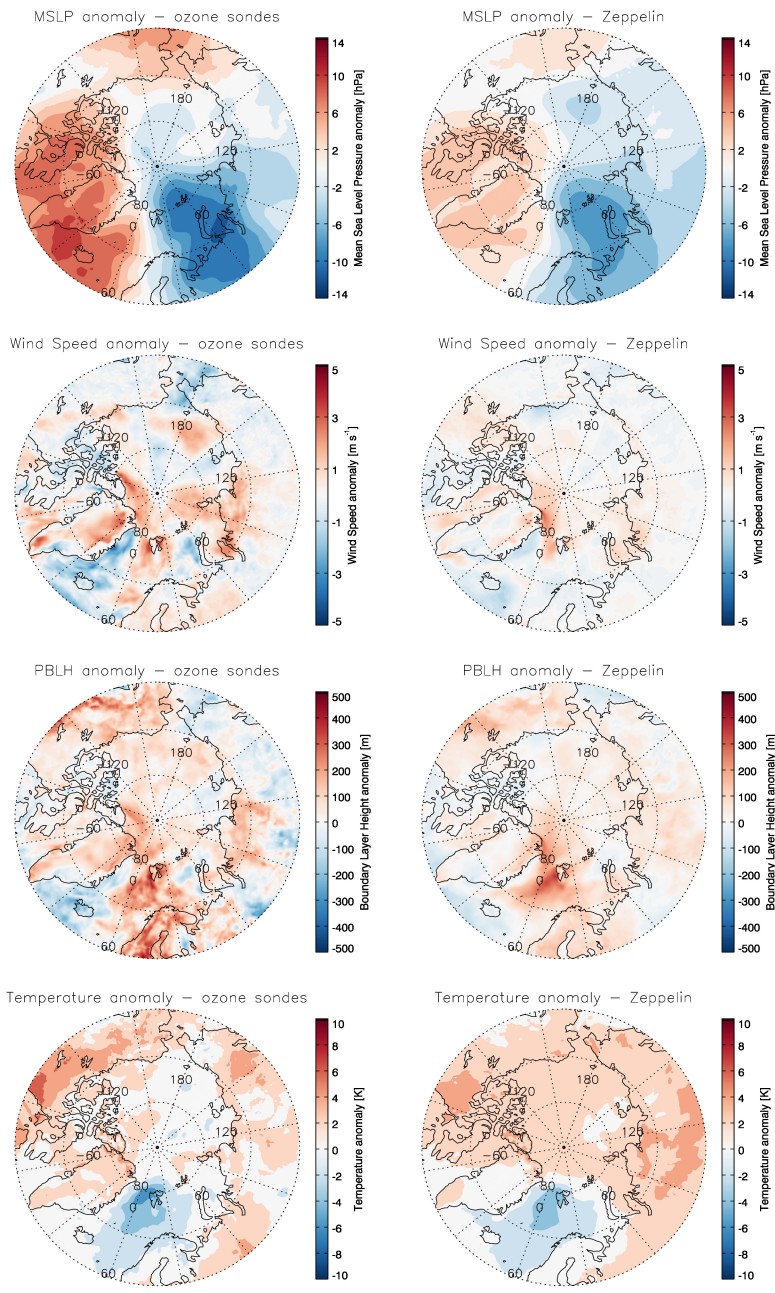

**Figure 3.** MSLP, wind speed, PBLH, and temperature anomalies for ODE and no ODE data points using a ozone threshold value of 15 ppb, based on the ozone sonde data set (left column) and Zeppelin data set (right column).





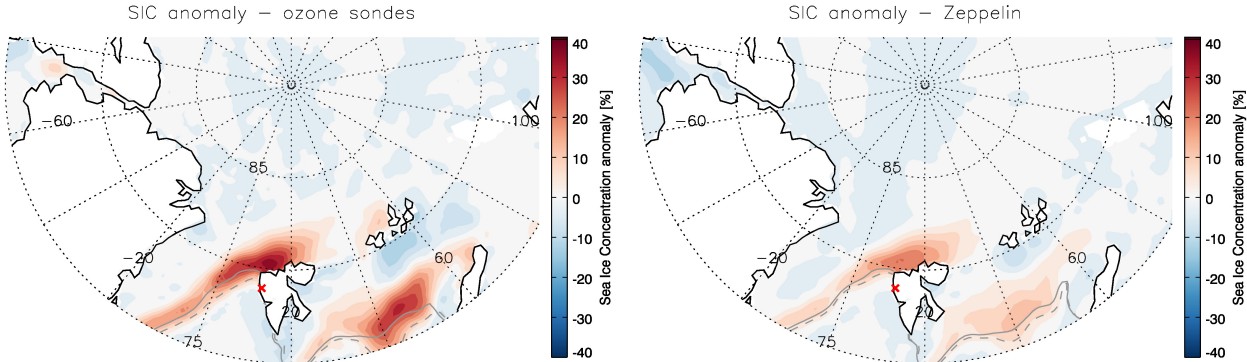

**Figure 4.** SIC anomalies for ODE and no ODE data points, based on the ozone sonde data set (left) and Zeppelin data set (right). The red cross marks the location of Ny-Ålesund. The solid grey line marks the mean SIE for no ODE data points and the dashed grey line for ODE data points.

towards Svalbard. This causes an elevated, unstable PBLH, with higher wind speed and presumably blowing snow where Br can be recycled and transported to Ny-Ålesund, depleting ozone on its way and in Ny-Ålesund.

## 3.2 Sensitivity analysis

### 3.2.1 Analysis of different threshold values

In this section, the influence of the ozone threshold on the composite analysis is discussed. The same method as above is applied, but with a threshold of 20 ppb instead of 15 ppb. This leads to 29 ODE and 228 no ODE days in the ozone sonde data set (difference to the amount of measurements at 15 ppb: 15) and 2157 ODE and 23489 no ODE hours for the Zeppelin data set (difference to the amount of measurements at 15 ppb: 920). The results for BrO and the meteorological conditions are shown in Figure 5 for the Zeppelin data set and in the appendix Figure B1 for the ozone sonde data set. Compared to the results for 15 ppb shown in Figure 2 for BrO and Figure 3 for the meteorological conditions, only minor differences are noticeable for the Zeppelin data set. The BrO anomaly in the 20 ppb data set north of Svalbard is slightly less pronounced than in the 15 ppb data set. The same pattern is also noticeable in the MSLP at the lower pressure area east of Svalbard and in the PBLH west of Svalbard. The wind and temperature anomalies are almost identical in the area around Svalbard.

For the ozone sonde data set, the differences between the results when using the 15 ppb threshold shown in Figure 2 for BrO and Figure 3 for the meteorological conditions and the 20 ppb shown in Figure B1 are more pronounced compared to the Zeppelin data set. Again, the BrO anomalies, as well as the low pressure east of Svalbard and the PBLH anomalies are weaker in the 20 ppb data set as well as the wind speed. The temperature anomaly around Svalbard is very similar for both ozone thresholds. However, it is warmer throughout the Arctic region at ODEs for the 15 ppb threshold. This may be explained by the increase of the number of ODEs in March when using the 20 ppb threshold, where the temperature in the Arctic is lower




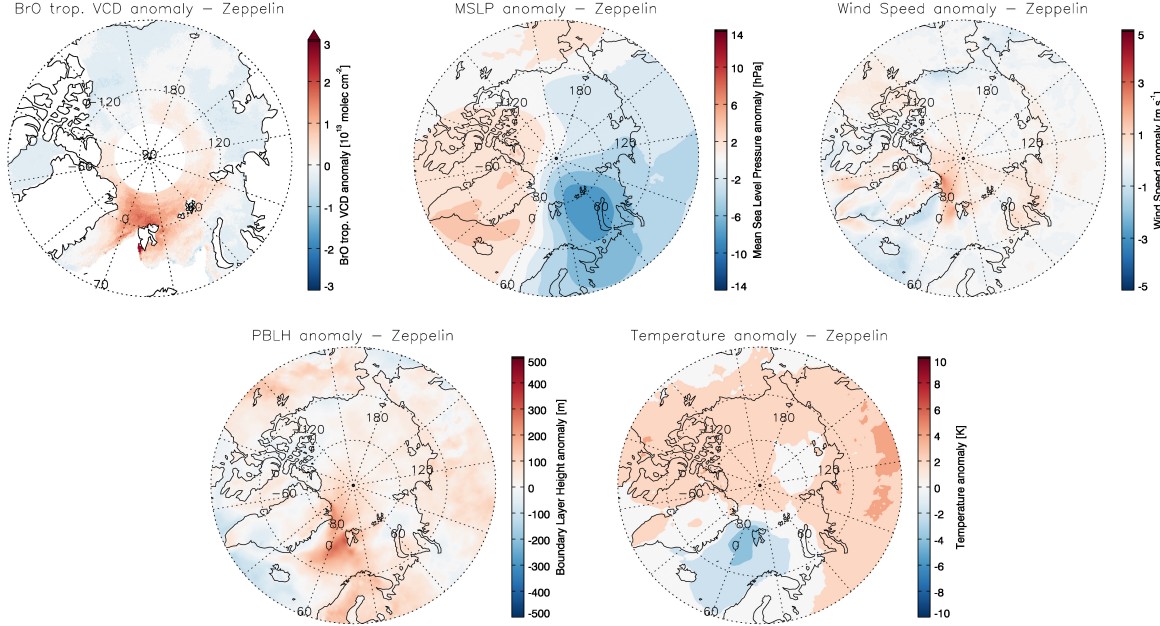

**Figure 5.** BrO, MSLP, wind speed, PBLH, and temperature anomalies based on the Zeppelin data set using a ozone threshold value of 20 ppb.

than in May.

Overall, the anomalies are slightly less pronounced in both data sets when using the 20 ppb threshold. But no major differences in BrO and meteorological anomalies are observed when changing the ozone threshold value.

### 3.2.2 Analysis of anomalies by month

To analyse the influence of seasonal effects, March, April and May data were analysed separately between 2010 and 2021 for both data sets. Figure 6 shows the results for the Zeppelin data set. The results for the ozone sonde data set are shown in the appendix Figure C1. In the BrO anomalies based on the Zeppelin data set, a decrease of BrO is visible between March and May. Large anomalies are visible in March mainly north of Svalbard. In April, the BrO anomalies are still positive, but not as pronounced and more homogeneously distributed over the Arctic ocean. The anomalies in May are smaller compared to the other two months. This is unexpected, since the largest number of ODEs has been observed in May on Zeppelin mountain (see Table 2). In general, it is expected that with increasing temperature at the end of spring the amount of inorganic $Br_2$ released from the cyrosphere decreases, and therefore less Br is available to deplete ozone. In the ozone sonde data set, three ODEs are observed in May (see Table 1), and the BrO anomalies in Figure C1 show enhanced BrO values are observed throughout the Arctic region for these two days. One possible explanation might be that only a few observed ODEs were induced by Br transported to Ny-Ålesund and mainly locally induced ODEs with low wind speed and stable boundary layer were observed





on Zeppelin mountain. A decrease of the anomaly can also be seen in the MSLP. From March until May, the area of the lower pressure anomaly becomes weaker and smaller. This decrease is connected to the wind speed anomaly, which also decreases, especially north-east of Greenland and west of Svalbard. The temperature anomaly around Svalbard also decreases.

Interestingly, in May the region around the North Pole seems to be slightly cooler during ODEs, which is not observed in March or April. Overall the observed trends towards a decrease of the described anomalies support the assumption that by the end of spring, ODEs in Ny-Ålesund might be more often induced locally and not due to transport from the pole region. The PBLH anomalies do not show any clear trends for these three months. The enhanced values north-east of Greenland in March and April are probably connected to the enhanced wind speed anomalies and support the assumption, that air is transported

from this area to Svalbard.

### 3.2.3 Analysis of anomalies by year

In order to analyse the influence of each year on the results of the composite analysis, a yearly evaluation was performed using the Zeppelin data set. The ozone sonde data set was omitted here because only in 4 of the analysed years, more than one ODE was measured (see Table1), which does not lead to robust results. Note that the year 2018 is not included in the analysis

because there were no ODEs measured that year (see Table 2). The appendix Figure D1 shows the mean BrO anomalies for each year. The years 2016 and 2020 show very high anomalies of BrO north of Svalbard. Except for 2012 and 2015, the other years show enhanced values around the Svalbard region as well, albeit not as strongly as the two previously mentioned years. Enhanced values in 2012 are mainly observed in the Fram Strait area, but also directly at the west coast of Svalbard over Ny-Ålesund. In 2015, enhanced BrO values are visible north-east and east of Greenland and north-east of Svalbard. Overall,

each year shows positive BrO anomalies during ODEs, but the years 2016 and 2020 might have the greatest influence on the composite analysis.

The MSLP plots in Figure D2 show very strong pressure anomalies with lower pressure east and higher pressure west of Svalbard in the years 2011, 2015, 2016, and 2020. In 2012, 2014 and 2019, a lower/higher pressure dipole is still visible even though not as clearly. For 2017 and 2021, there is overall a lower pressure east and west of Svalbard, but with a more

pronounced lower pressure anomalies on the east side, especially in 2021. No dipole is visible in 2010 and 2013 but only a lower pressure area north of Svalbard. It can be concluded, that a lower pressure east and/or a higher pressure west of Svalbard occurs during ODEs in most years, leading to a transport of cold polar air from the north to Svalbard.

These observations are consistent with the results of the wind speed plotted in Figure D3. In 2010, where no MSLP anomaly dipole was observed, negative wind speed anomalies are obtained. It is the only year where lower wind speeds are observed

during ODEs. Additionally, it is the year with the smallest number of observed ODEs, other than 2018. Besides 2011 and 2014, which show hardly any change in wind speed close to Ny-Ålesund during ODEs, for every other year, increased wind speeds are observed. For the years with positive wind speed anomalies it is very likely that recycled Br and/or ozone depleted air is transported to Ny-Ålesund, leading to the ODEs. In the areas of enhanced wind speed anomalies, increased PBLH anomalies are visible in Figure D4 as well.

The temperature anomalies for each year in Figure D5 consistently show lower temperatures in the area west of Svalbard





**Figure 6.** MSLP, wind speed, PBLH, and temperature anomalies based on the Zeppelin data set for each month: March (left column), April (middle column), and May (right column).





during ODEs. This finding supports the assumption that lower temperatures accelerate the $Br_2$ release from the cryosphere and therefore can deplete ozone in a sunlight atmosphere. In addition, a seasonality of the temperature is observed in the plots. In years where most of the ODEs took place in May, the Arctic region shows overall positive temperature anomalies, except for the region around Svalbard. On the other hand, in years such as 2016 where most of the ODEs occurred in March, a continuous
negative temperature anomaly is observed, due to the lower temperatures in March compared to May.

Overall, there is a strong inter-annual variability. Some years have low numbers of ODEs compared to others but show strong anomalies (e.g. 2016), while the year with the most ODEs (e.g. 2014), shows weaker anomalies. In 2018, no ODEs were measured in either dataset. Several years including 2015, 2016, and 2020 show strong anomalies, which significantly influence the composite analysis presented in section 3.1. However, other years still show a similar pattern of anomaly, albeit in a
weakened form. Some years, such as 2010 and 2013, show different anomaly patterns in terms of MSLP and wind, but still have lower temperature values on ODE days. It might be possible that in years with ODEs in late spring, where no strong MSLP dipole is visible, ODEs might be induced due to a temperature drop that leads to the formation of new sea ice in the area very close to Ny-Ålesund (e.g. Kongsfjorden), where $Br_2$ is emitted locally and initiate the ozone depletion cycle. Another possibility could be sea spray, which infuses coastal snow with salty aerosols and a subsequent temperature drop can then
initiate the bromine reaction cycle.

## 3.3   ODE Case Study

At the beginning of April 2020, a major ODE was observed in the ozone sonde observations and in the Zeppelin in-situ measurements. The event started early in the day of April 2nd and lasted for two days until the end of April 3rd (see Figure 7). During these two days, ozone was depleted below the detection limit, the most severe event recorded between 2010 and 2021.
This event occurred in connection with a high-latitude cyclone, a situation which was already discussed in other studies (e.g., Blechschmidt et al., 2016; Chen et al., 2022; Choi et al., 2018). Before the onset of the ODE in Ny-Ålesund, a large comma shaped BrO plume was observed north of Svalbard in the TROPOMI satellite data on April 1st (see upper left Figure 9). The plume arrived in Ny-Ålesund in the afternoon, as recorded by the stationary MAX-DOAS instrument. As shown in the blue line in the left Figure 8, a BrO peak is observed in the same altitude region where a decrease in ozone can be identified. For
these three days, daily ozone sonde data are available. Ozone was depleted below detection limit in the Zeppelin data for the two following days, as well as in the sonde data up to around 1.5 km altitude. Enhanced BrO values were also detected by the MAX-DOAS instrument in the same altitude region (see Figure 8). For April 2nd, the TROPOMI satellite image (see upper middle Figure 9) shows several BrO hotspots scattered over Svalbard as well as north-east of Greenland. For the following day, the main plume is partly visible east of Greenland and only a very weak signal of BrO is visible over Svalbard (see upper right
Figure 9). However, the MAX-DOAS instrument (see Figure 8 right) shows elevated BrO values for that day and ozone is still depleted below the detection limit of the sonde, which leads to the assumption that the amount of BrO is not fully captured by the satellite observations.





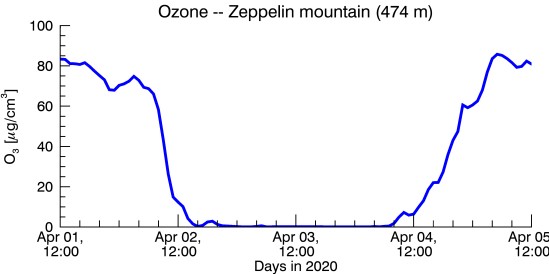

**Figure 7.** Zeppelin ozone data between April 1st at 12:00 UTC and April 5th at 12:00 UTC capturing a major ODE, where ozone is depleted below detection limit.

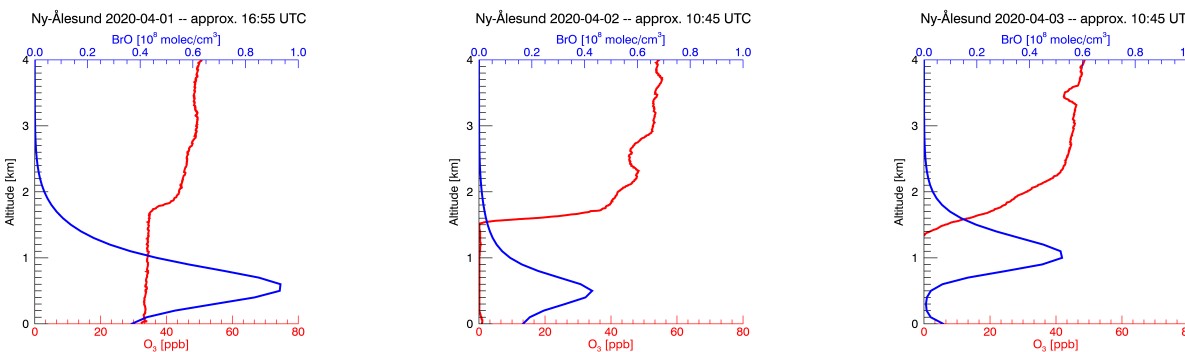

**Figure 8.** Ozone sonde measurements (red) together with BrO MAX-DOAS profiles (blue) for (left) April 1st, (middle) April 2nd, and (right) April 3rd, 2020.

To investigate the meteorological conditions during this event, WRF simulations were conducted as shown in Figure 9. As
in the composite analysis, mean sea level pressure, wind speed, planetary boundary layer height, and temperature are plotted
here. For better comparison, the WRF output closest in time to a TROPOMI overpass over Svalbard is used. The simulations
show a low pressure system located northeast of Svalbard on April 1st that propagates southwest towards Svalbard arriving on
April 2nd. The system then moves over Svalbard and progresses further south on April 3rd. The low pressure system in combi-
nation with the high altitude ozone depletion captured by the ozone sondes corresponds with the findings of Jones et al. (2010),
who observed ozone depletion above 1 km in conjunction with a low pressure system and high wind speed in the Southern
hemisphere. Together with the low pressure system, there are increased wind speeds at the front of the cyclone, transporting
cold polar air southwards. The lifting of the planetary boundary layer at the front of the cyclone can be seen as well. These
findings are consistent with the observations in Chen et al. (2022), where it is assumed that the observed BEE in Ny-Ålesund
was primarily initiated by blowing snow, presumably brine covered, triggered by a polar cyclone approaching Svalbard.




**Figure 9.** (first row) Tropospheric BrO VCD for three satellite overpasses over Svalbard on (left) April 1st, (middle) April 2nd, and (right) April 3rd. The images below show the WRF output closest in time to the satellite images for (second row) MSLP, (third row) wind speed, (fourth row) PBLH, and (fifth row) temperature.



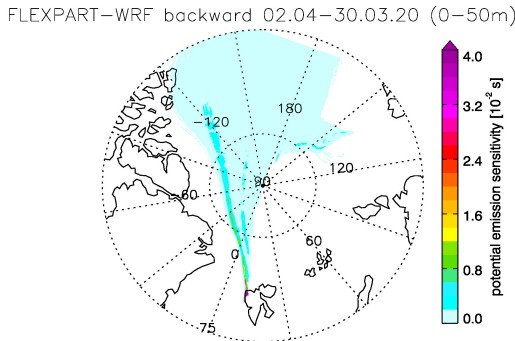 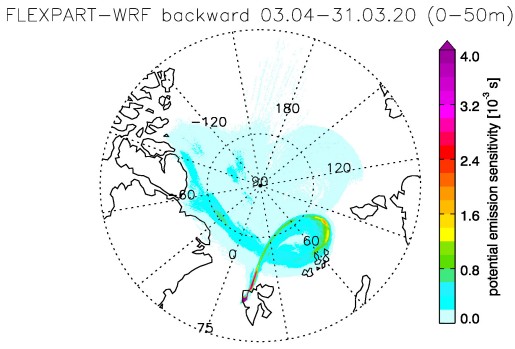

**Figure 10.** Output of two FLEXPART-WRF backward runs initialized on (left) April 2nd at 11:00 UTC and (right) April 3rd at 11:00 UTC.

Two FLEXPART-WRF backward runs were executed to investigate the origin of the air masses arriving in Ny-Ålesund. The first run was initialized for April 2nd and the particles were traced back three days (see Figure 10). Due to the assumption that the source of Br is salty sea ice and therefore close to the ground, only the trace of the particles within 50 m above sea level is shown here. As can be seen, the particles arriving at Ny-Ålesund originate from the north with the highest sensitivity and

therefore possible source close to Ny-Ålesund. Note that the particles released in the run can represent either Br that depletes ozone or air masses where ozone is already depleted. The ODE could have been induced by local Br or Br that was recycled over the Arctic sea ice and transported to Ny-Ålesund. As another option, ozone depleted air from the Arctic sea ice arriving at Ny-Ålesund led to the ODE. The second run initialized on April 3rd shows a transport route of the particles that follows the shape of the polar cyclone. It is plausible that on this track a lot of ozone depletion occurred and the air arriving in Ny-Ålesund

already contained very little ozone.

This case study nicely illustrates the ODE conditions found by the results of the composite analysis. A low pressure system from the north east approaching Svalbard leading to high wind speeds and low temperatures transported from the north to Ny-Ålesund resulting in increased BrO values and a severe ODE.

## 4 Summary and conclusion

This study investigated the meteorological conditions that are associated with ODEs occurring in Ny-Ålesund during the period 2010 – 2021. A composite analysis was performed using two different ozone data sets: the ozone sonde record and the hourly in-situ observations at the Zeppelin observatory. The composite analysis showed that ODEs develop in an atmospheric blocking situation, with a low pressure anomaly located over the Barents Sea and anomalously high pressure in the region of the Icelandic low. This leads to the transport of cold polar air from the north to Ny-Ålesund, with higher wind speeds potentially inducing

blowing snow along the way. The PBLH is anomalously high in the areas of higher wind speed. In addition to the pressure and wind anomalies, lower than normal temperatures are found around Svalbard during ODEs as well as enhanced BrO columns. Since the coastal area around Ny-Ålesund and Kongsfjorden is mostly sea ice free in spring nowadays, air depleted in ozone



and enriched in BrO is probably transported to Ny-Ålesund. During the long range transport over the sea ice covered areas
north of Svalbard, recycling of bromine on blowing snow is possible. Analysis of the days before and after an ODE showed

the development of a polar cyclone with the highest anomalies in pressure, wind speeds, PBLH and temperature on the day of
the ODE and decreasing anomalies thereafter.

To analyse the robustness of the results, both data sets were additionally evaluated with a higher ozone threshold value of 20
ppb. For both datasets, the anomalies were slightly less pronounced at the 20 ppb threshold, with the differences in the ozone
sonde dataset being slightly larger than in the Zeppelin dataset. It is assumed that with weaker weather systems, the ODEs in

Ny-Ålesund also become less pronounced.

The monthly analysis showed a decrease of the anomalies towards May in the Zeppelin data set. This leads to the assumption
that in early and mid spring, ODEs near Ny-Ålesund are induced more often by low pressure systems with high wind speeds
and cold air masses transported from the north. In late spring (around May), ODEs are probably more often induced during
calm meteorological conditions where bromine can accumulate and deplete ozone. Possible sources could be on the one hand

sea spray that infuses coastal snow with sea salt aerosols which, at a sufficiently low temperature, releases bromine to the
troposphere. Another option might be that due to a temperature drop, fresh sea ice is formed locally serving as bromine source.
Alternatively, ozone depleted air from the north can still be transported to Svalbard leading to ODEs without local ozone
depletion chemistry.

The annual evaluation shows that several individual years present the same pattern of anomalies as the combined data set.

These years have the strongest influence on the overall outcome of the composite analysis. Some years show only weak
anomaly patterns or have a different anomaly pattern. This applies especially for MSLP, wind and PBLH. However, all years
have in common that there are lower temperatures in Ny-Ålesund during ODEs as well as higher BrO columns upwind of the
station.

Finally, a case study in the beginning of April 2020 was introduced when ozone was depleted below detection limit for two days.

The ODE occurred in conjunction with a low pressure system arriving at Svalbard from the north-east with high wind speeds
and transport of cold polar air from the north. It showed similar meteorological patterns as the anomalies of the composite
analysis and is an example of how these conditions can lead to severe ODEs in Ny-Ålesund. It is very likely that most of the
ozone poor air and BrO was transported to Ny-Ålesund, with recycling of BrO over the sea ice region. In summary, the majority
of ODEs in Ny-Alesund appear to be driven by meteorological conditions favouring transport of cold polar air to Svalbard, in

particular in early spring. With regard to the rising temperatures in the Arctic (Serreze and Barry, 2011), the consequences of
changes in the meteorological parameters and SIC due to rising temperatures on halogen release from the cryosphere requires
further investigation. Due to a decrease of the sea ice extend, less source area for BEE will be available. The evolution of the
amount and strength of BEEs and and consequently ODEs under the aspect of changing meteorological drivers will be a future
topic to discuss.



*Code availability.* WRF source code from the National Center for Atmospheric Research (NCAR) Version 4.2 was accessed via http://www2.mmm.ucar.edu/wrf/users/ (Skamarock et al., 2019). Flexpart-WRF was obtained from their website https://flexpart.eu.

*Data availability.* Ozone sonde records are available from the Network for the Detection of Atmospheric Composition Change (NDACC) at https://www.ndacc.org. Ozone in-situ data from Zeppelin mountain were provided by the Norwegian Air Research Institute (https://ebas-data.nilu.no/) (Platt et al., 2022). ERA5 meteorological and sea ice data is available at ECMWF (Hersbach et al., 2020). EASE-Grid Sea

ice Age version 4 were accessed via https://nsidc.org/data/nsidc-0611/versions/4. NCEP FNL data was provided by the Computational and Information Systems Laboratory (CISL) on their website http://dss.ucar.edu/. The long term BrO data set from Bougoudis et al. (2020) is available through the World Data Center PANGAEA (https://doi. pangaea.de/10.1594/PANGAEA.906046





# Appendix A

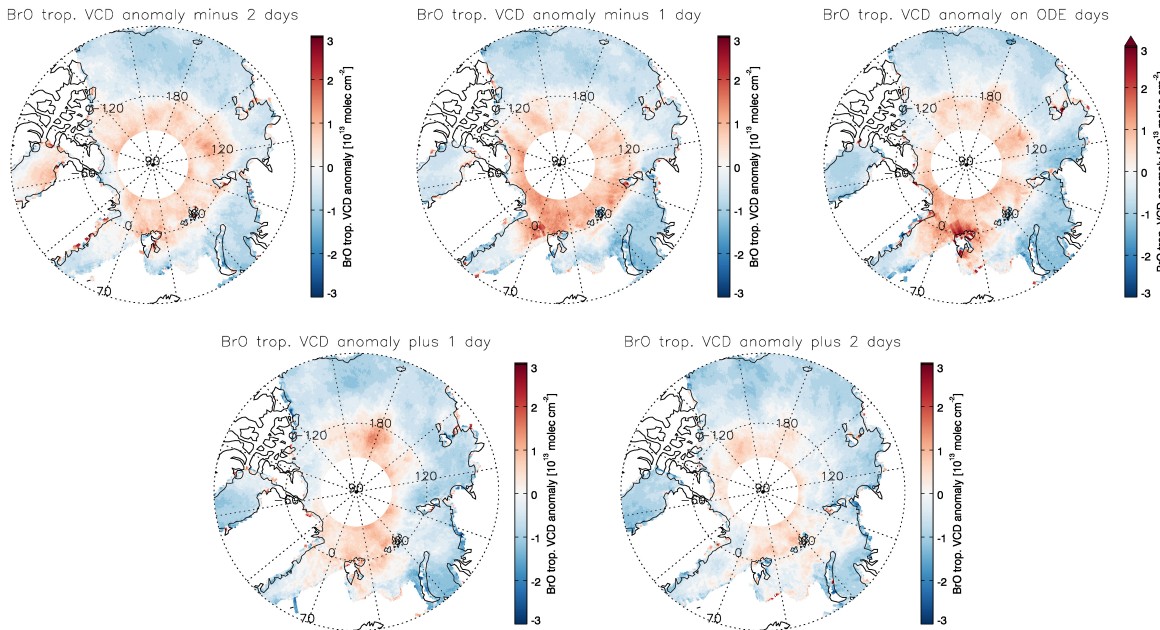

**Figure A1.** Tropospheric BrO VCD anomalies one and two days before and after the ODE as well as the day of the ODE based on the ozone sonde data set.




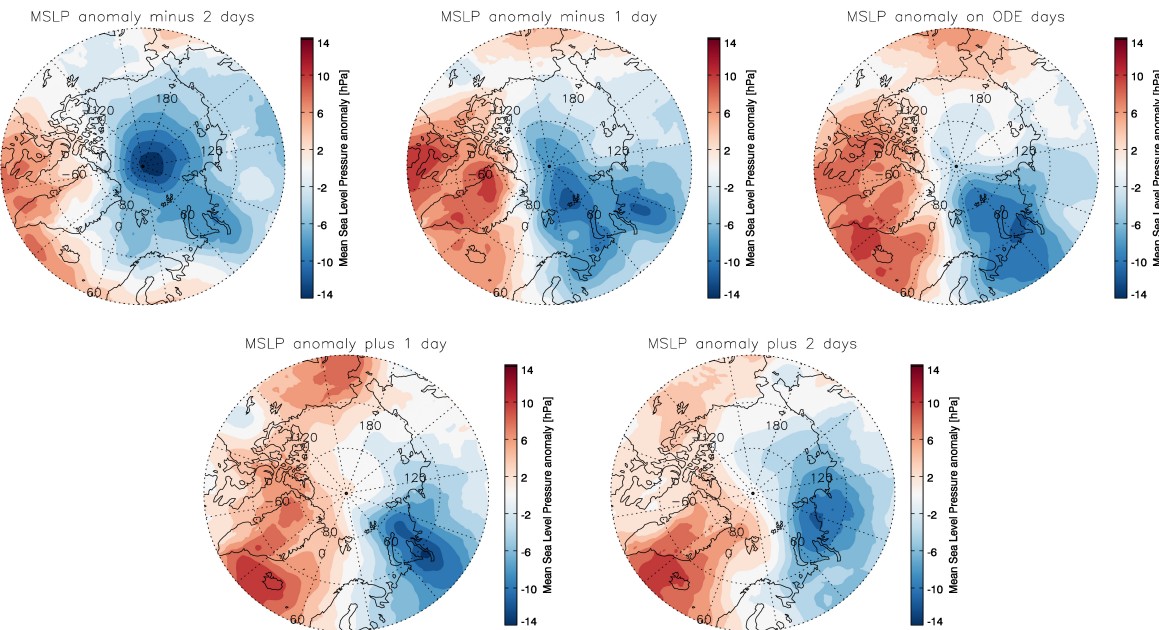

**Figure A2.** MSLP anomalies one and two days before and after the ODE as well as the day of the ODE based on the ozone sonde data set.

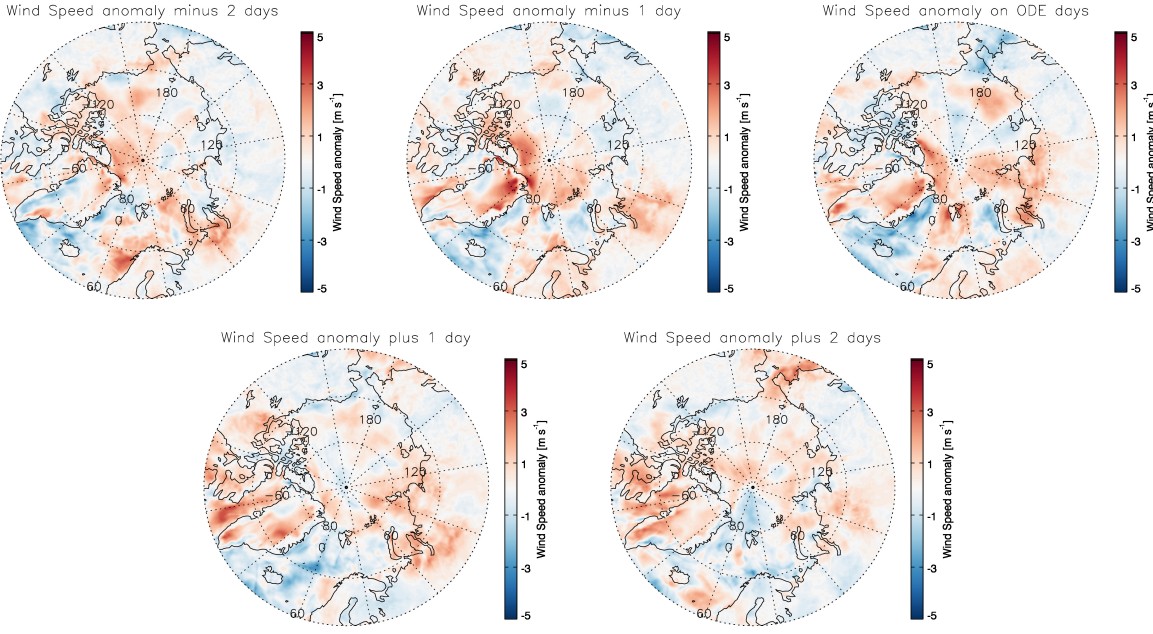

**Figure A3.** Wind speed anomalies one and two days before and after the ODE as well as the day of the ODE based on the ozone sonde data set.



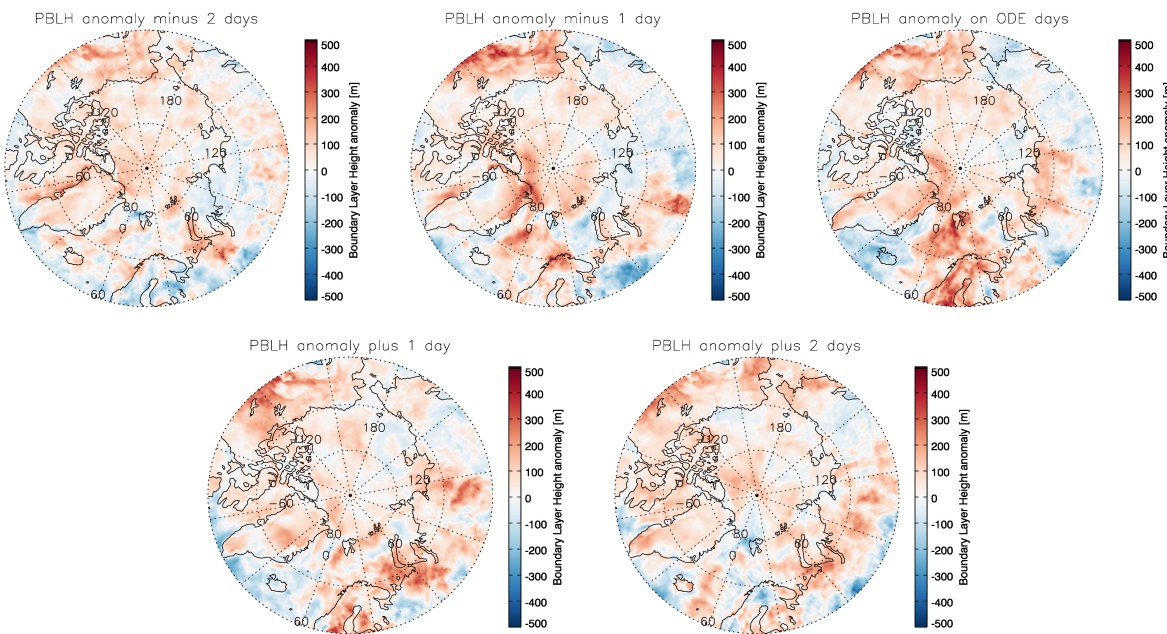

**Figure A4.** PBLH anomalies one and two days before and after the ODE as well as the day of the ODE based on the ozone sonde data set.

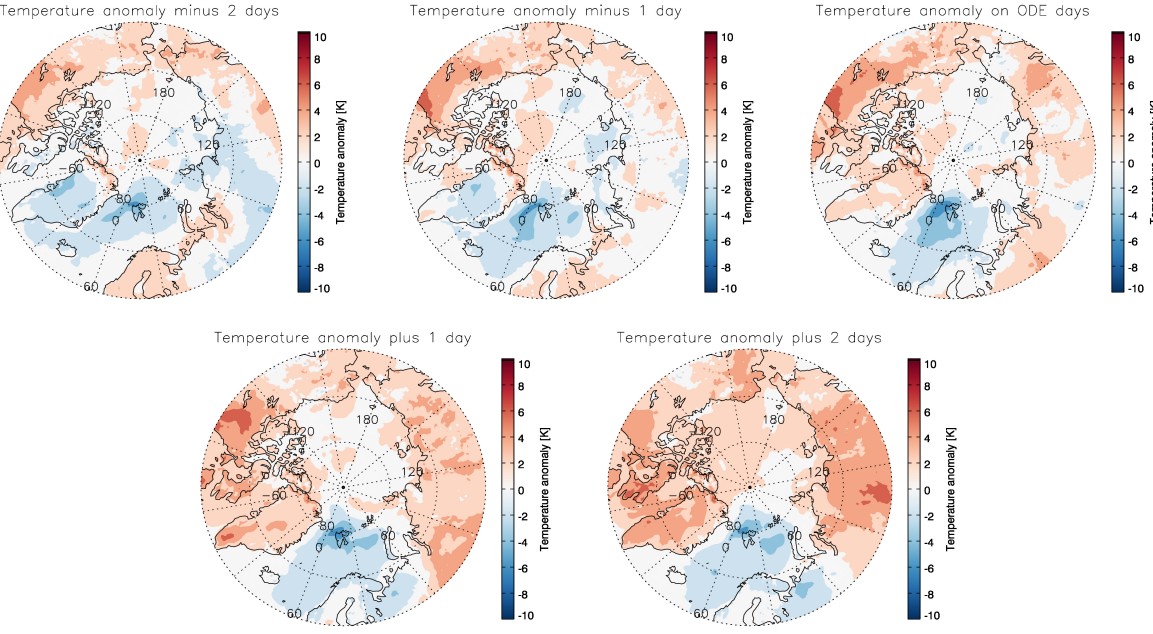

**Figure A5.** Temperature anomalies one and two days before and after the ODE as well as the day of the ODE based on the ozone sonde data set.





# Appendix B

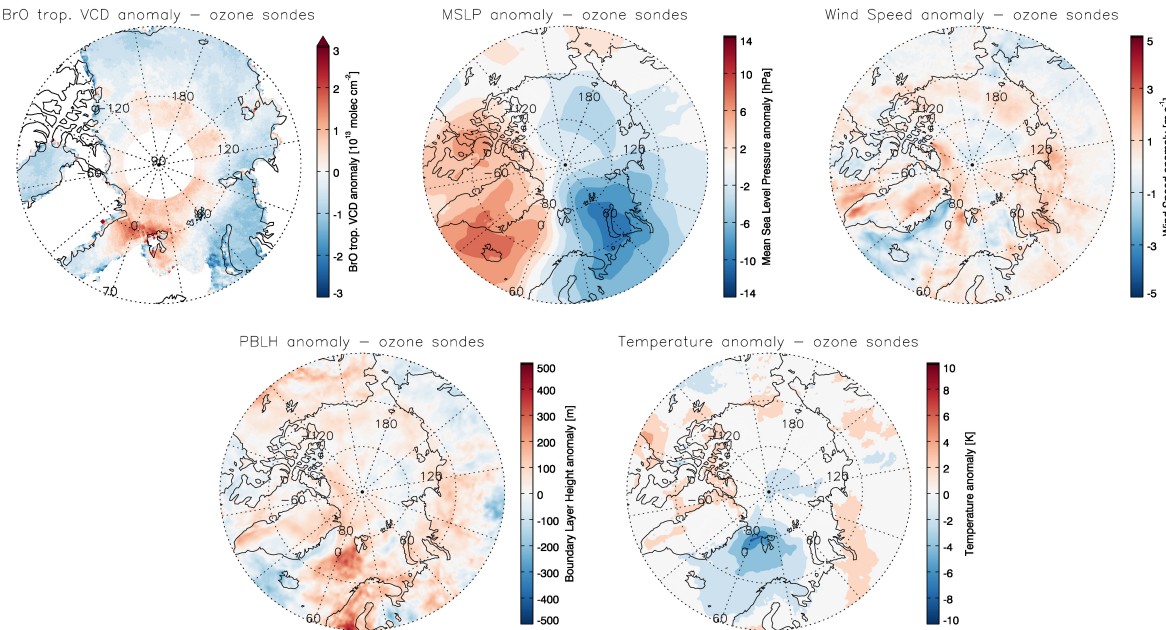

**Figure B1.** BrO, MSLP, wind speed, PBLH, and temperature anomalies based on the ozone sonde data set using a ozone threshold value of 20 ppb.



**Appendix C**





**Figure C1.** MSLP, wind speed, PBLH, and temperature anomalies based on the ozone sonde data set for each month: (left column) March, (middle column) April, and (right column) May.



## Appendix D

**Figure D1.** Tropospheric BrO VCD anomalies based on the Zeppelin data set for each year between 2010 and 2021. No ODEs were measured in 2018, and therefore no anomalies are calculated.





**Figure D2.** MSLP anomalies based on the Zeppelin data set for each year between 2010 and 2021. No ODEs were measured in 2018, and therefore no anomalies are calculated.





**Figure D3.** Wind speed anomalies based on the Zeppelin data set for each year between 2010 and 2021. No ODEs were measured in 2018, and therefore no anomalies are calculated.



**Figure D4.** PBLH anomalies based on the Zeppelin data set for each year between 2010 and 2021. No ODEs were measured in 2018, and therefore no anomalies are calculated.





**Figure D5.** Temperature anomalies based on the Zeppelin data set for each year between 2010 and 2021. No ODEs were measured in 2018, and therefore no anomalies are calculated.



*Author contributions.* This study was designed by BZ, AR, AB. BZ performed the data analysis and wrote the manuscript with contribution from AR, TB, PvdG and JB. Data input was provided from PvG, TB, IB, and SS. All authors contributed to the final manuscript.

*Competing interests.* The authors declare that they have no conflict of interest.

*Acknowledgements.* This study was partly funded by the Deutsche Forschungsgemeinschaft (DFG, German Research Foundation) – project no. 268020496 – TRR 172, within the Transregional Collaborative Research Center "ArctiC Amplification: Climate Relevant Atmospheric and SurfaCe Processes, and Feedback Mechanisms (AC)[3]". Thanks to the developers of WRF and FLEXPART-WRF for the model source codes that are available on their webpages. Furthermore, we thank ECMWF for providing their ECMWF ERA5 reanalysis products on their website. We thank the team behind the Zeppelin Observatory and EBAS for making their ozone data available on their websites. The

EASE-Grid Sea Ice Age version 4 was thankfully provided by NSIDC. Parts of the sea ice evaluation were kindly supported by Lars Aue.



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
