# Peer review of "Investigation of meteorological conditions and BrO during Ozone Depletion Events in Ny-Ålesund between 2010 and 2021"

_EGUsphere, 2023_

## Referee Comment (RC2)

**Review egusphere-2023-522**

Investigation of meteorological conditions and BrO during Ozone Depletion Events in Ny-Ålesund between 2010 and 2021

**1 general comments**

`evaluating the overall quality of the discussion paper`
The paper investigates the importance of polar cyclones and associated meteorological condition on ozone depleting events (ODEs) observed in the proximity of Ny-Ålesund. The authors apply complementary datasets of both ozone ($O_3$) and BrO (e.g. in situ observations and various remote sensing products) and combine these with Lagrangian (FLEXPART) and Eulerian (WRF) modeling techniques. Though, no Chemistry Transport Model (CTM) or process modeling were applied in this study. Overall, the paper is

- well-structured,
- comprehensible in its application of the composite data analysis.

The language is

- comprehensive
- and for most parts only minor corrections may apply.

Though this paper was, overall, a real pleasure to read, a major shortcoming emerges from the distinction between background (no ODE) state of the atmosphere and ODE state. The authors define *no ODE*, in principle, by $Y_{\mathrm{noODE}} = Y_{\mathrm{tot}} - Y_{\mathrm{gap\,O_3\,data}} - Y_{[O_3]<15\,\mathrm{ppb}}$, where $Y$ in this case refers to one record in 1-hourly meteorological data (e.g. $Y_{\mathrm{tot}}$ would correspond to 744 records in March). Applying this definition of the background state of the atmosphere most likely reduces the explanatory power of the anomaly analysis because ODE conditions are likely to be contained in the *no ODE* data records. This clearly emerges from the sensitivity analysis in Section 3.2.1 where a lower threshold (20 ppb) was applied. This weakened the found dipole structure. The anomaly analysis could potentially benefit from stronger constraints on the background state, however, the general outcome of the study will probably not change significantly. Hence, either the authors, unconstrained by computational and human resources, repeat the anomaly analysis with a more clear-cut background state or discuss the implications more thoroughly in Section 3.2.1.

**2 specific comments**

`individual scientific questions/issues`

- p4 l116: "[...] not necessarily capture all ODEs at sea level.": Could you possibly give an estimate based on your composite analysis? Or give a number of coincident events?

- p5 l121: "[...] ODEs occur mainly in polar spring [...]": Is there evidence that they do occur in, e.g., fall?
- p5 Table 1: Would it make more sense to differentiate between ODEs and total number of ascents in the respective month?
- p6 Table 2 (and respective paragraphs in the text): How is "ODE" count exactly defined here? It seems as if you count consecutive hours of below threshold $O_3$ as one event each. An ODE can, in fact, as your correctly wrote and showed (Fig 7), last for several days. Hence, your definition of ODE in this Table is not consistent with, e.g., Section 3.3. In any case, no ODE and ODE [hours with $O_3$ below threshold] do not add up to the total number of hours in each month, e.g., 744 h in March. It is not clear if this is caused by gaps in data. Could you clarify this?
- p5 l122: How did you define the 15 ppb threshold? Is it possible to clearly separate populations in a histogram of $O_3$ monitoring (bi-modal distribution)?
- p5 l129: How like do you detect "fake" ODEs due to low hemispheric $O_3$ background especially at higher thresholds?
- p7 l154–177: "From 2010 until 2017 [...]": Too specific and technical. You may summarize the technical information about the different satellite products in a table and keep only additional information in the text.
- p8 l195–196: "[...] two domains were used in a two-way nested run, i.e. the values of the coarse domain are overwritten by values of the higher resolution [...]": That's not entirely correct, as changes in the inner and outer domain influence field information of one another in a two-way coupled model system. Either explain the coupling in detail or drop the second half of the sentence starting with "i.e.".
- p9 l219-223; eq (1): $Y_{ODEanom} = \overline{Y}_{ODE} - \overline{Y}_{noODE}$: This method is slightly problematic in the sense that you may have erroneously detected ODEs / noODEs in both data sets (even more so in the data sets with the lowered threshold of 20 ppb). These will blur the signal you are after. In this context, the definition of ODE that you apply to the ozone monitoring data is not sound (see also comment regarding Table 2). Consecutive hours of below threshold ozone concentrations are a necessity to identify an "event", hence the number of hours of ODE is misleading with respect to ODE statistics. The lag analysis that you apply in the following should be associated to the onset/end of an individual ODE. In summary, if you use times of "no ODE" you have to make sure that these really **do not contain any ODE** which I currently cannot see satisfied in your analysis. Another threshold, e.g. for *normal* ozone concentrations, could do the trick.
- p10 Fig 2: Regarding the shown data, where does the "white" area south of Svalbard come from? Did you use some kind of "sea ice edge" filtering to exclude these data or are there no data due to retrieval constraints?
- p11 l274: "long computing times": On which kind of system? Personal computer, HPC? If former is the case an application of other computational resources should have been considered for this analysis, perhaps?

- p13 Section 3.2.1: How did you derive the different threshold values? Are they "random" choices or were the derived from a bi-modal analysis of ozone concentration distributions if that is even possible? As you show in this section, the choice of threshold is crucial for identifying a causal signal. Have you analyzed coincident ODEs in both sounding and monitoring data with respect to the meteorological and BrO conditions? Or are there too few coincident ODEs?

- p14 l309: "Overall, the anomalies are slightly less pronounced when using the 20 ppb threshold.": This doesn't really come as surprise. By lowering the threshold you'd allow for more "false positive" ODEs that can originate from both transport and mixing of air parcels with different trace gas concentrations, as well as the inclusion of subsiding ODEs where the actual cause is not present in your time lag analysis.

- p17 l361–370: "Several years [...] show strong anomalies [...] other years still show similar patterns [...]": Let's turn this around: If you'd find similar patterns in your meteorological fields but no ODE, wouldn't that mean that these meteorological conditions do not suffice as cause of ODEs?

- p17 l385–387: "[...] the amount of BrO is not fully captured by the satellite observations." Could this be due to the algorithm used to separate tropospheric and stratospheric columns?

- p21 l452: "Due to a decrease of sea ice extend less source area for BEE will be available." Without taking the processes associated with BEE and ODE into account, this remains highly speculative. Given blowing snow on first year sea ice and brine on (young) sea ice is among the major sources of Br in the polar spring boundary layer (see e.g. `10.5194/acp-12-6237-2012` for a review of processes), the extent of the sea ice is probably less important compared to the structure and dynamics of the sea ice, higher wind speeds, and a change in frequency in the occurrence of polar lows. Dynamics of sea ice formation have been notoriously hard to detect with passive sounding satellite-born instruments, but advances might have been made in recent years.

**3 technical corrections**

`purely technical corrections`

- p2 l24: "during sunlight": term?
- p2 R1–R6: Typesetting of chemical formulas: $Br_2 \rightarrow \text{Br}_2$, asf.
- p6 Fig 1: (left) Maybe indicate altitude of Zeppelin observatory?
- p7 l154 ff: "From 2010 until 2017 [...]": Duplicate of paragraph "To analyse [...]" (p7 l146 ff). Please condense the two paragraphs.
- p8 l183–185: "Additionally, to analyse [...]": This sentence might be grammatically incorrect. Maybe better: *Daily AMSR (...) sea ice concentration (SIC) observations on a $25 \times 25$ km$^2$ grid have been used to analyse the SIC [...].*

- p8 l184, l191, l190: Style! "was used" is used in each of these consecutive sentences. You may want to rephrase.
- p8 l196–198 and Fig 9: Definition of WRF domains: Which projections were used? Does the clipping shown in Fig 9 represent the boundaries of the outer domain? Would it be possible to indicate the location of the inner domain in the WRF related plots in Fig 9?
- p9 l213, l214, l225, and others: "time points" Incorrect term. Rephrase → *times*.
- p9 l215–216: "To separate the ozone data [...]": Repetition of Section 2.1. Remove or rephrase: *From the ozone data (see Section 2.1), we found 14 ODEs and 228 no ODEs in ozone sonde data records and [...].*
- p9 l219 ff: Typesetting of formulas. If not defined otherwise in the journal's style guide, non-indexing subscripts should not be set in italic font: $\overline{Y}_{ODE} \rightarrow \overline{Y}_{\text{ODE}}$.
- p10 Fig 2: "BrO VCD anomalies for ODE and no ODE" Caption text confusing? I would assume, you should drop "for ODE and no ODE" here. Regarding the shown data, were does the "white" area south of Svalbard come from? Did you use some kind of "sea ice edge" filtering to exclude these data or are there no data?
- p10 l235: Typo: "Ny-Alesund" → Ny-Ålesund
- p10 l242: Typo: "the the" remove one "the"; Missing comma after adverb?: "Normally the Icelandic low [...]"
- p10 l243: "Due to the lower pressure [...]": Perhaps *low pressure system*? (But I'm not firm in weather synoptics.)
- p15 l314–345: "[...] a more pronounced lower pressure anomalies [...]": Mismatching singular article and plural noun.
- p17 l378: "As shown in the blue line": preposition?: *As shown by the blue line*
- p17 l379: "in the same altitude": preposition?: *at the same altitude*
- p21 l452: "extend" → *extent*
- p27: Empty → Fig C1 should have appeared here.
- p25: Fig A5 is not referenced in the text.

---

## Author Comment (AC1)

**Comments from anonymous Referee #1:**

We would like to thank the reviewer for his/her helpful comments. We hope that we could address all questions and unclear points satisfactorily.

Legend: Author comments in blue, Referee comments in black.

**General Comments**

This manuscript investigates the meteorological conditions spatial distributions of BrO in the Arctic leading to ozone depletion events as observed by ozone sondes and an in situ ozone monitor in Ny-Alesund, Spitsbergen. This is done by separating the ozone time series into ODE and non-ODE periods using a threshold value, and by calculating maps of the anomaly of meteorological parameters, sea ice conditions and BrO VCDs over the Arctic for both situations. Based on these anomaly maps, the impact of the spatial distribution of temperature, wind speed, boundary layer height and pressure as well as BrO and sea ice coverage has been investigated. The manuscript confirms many findings from previous studies, such as the impact of polar cyclones on ozone depletion and the occurrence ODEs during high wind speeds, which are probably due to heterogeneous release of reactive bromine from saline aerosols. It is found that certain distributions of polar low- and high-pressure systems lead to the southward transport of ozone depleted air towards Ny-Alesund. Furthermore, the seasonal and inter-annual variation of these anomalies is investigated and a case study on a particular ODE is presented.

The meteorological conditions leading to a release of reactive bromine and a subsequent ozone depletion are still not fully understood. Therefore, this manuscript provides a valuable contribution to this field of research and fits well into the scope of ACP. The results of the study are described appropriately, but I feel that the description of the methods requires substantial revision. In particular, the "composite analysis" method presented in Sect. 2.7., which represents the key method of the study, should be re-written since it is lacking conciseness and is difficult to understand (see the specific comments below).

Following the suggestion of the reviewer, Section 2.7 has been rewritten (see below).

Sect. 3.2.1. describes in detail the impact of the ODE threshold values on the resulting anomaly maps, and concludes that there is only little impact on a qualitative basis. I therefore suggest to skip this section, and simply add the final sentence of this section ("No major differences in BrO and meteorological anomalies are observed when changing the ozone threshold value") to the methods section.

We have decided not to remove Section 3.2.1 as an additional discussion point was added there and as we think it is important to show the effects of different thresholds as these are chosen more or less randomly.

**Specific Comments**

P2, L47: In addition to chlorine, I think it would be worth mentioning iodine as a potential booster for ozone depletion (Benavent et al, 2022).

We agree that iodine could be relevant, and therefore already mentioned it further below in the text using the same reference (P3, L58-60).

P3, L68: Please add a reference for the lifetime of BrO and specify what exactly is meant with this value. While the photolytic lifetime of a BrO molecule is quite short, the lifetime of BrO in a certain air mass depends on various parameters, such as the presence of saline surfaces for recycling.

Following the suggestion of the reviewer, we have updated this sentence and added references:

*The photolytic lifetime of BrO is approximately one minute (e.g., Lehrer et al., 2004; Pratt et al., 2013)*

Here, we only refer to the photolytic lifetime of BrO. Although lifetimes are given in several publications (see below), we did not find a satisfactory reference describing how these values were determined.

Liao, J., et al. (2011), A comparison of Arctic BrO measurements by chemical ionization mass spectrometry and long path-differential optical absorption spectroscopy, *J. Geophys. Res.*, 116, D00R02, doi:10.1029/2010JD014788.

Lehrer, E., Hönninger, G., and Platt, U.: A one dimensional model study of the mechanism of halogen liberation and vertical transport in the polar troposphere, Atmos. Chem. Phys., 4, 2427–2440, https://doi.org/10.5194/acp-4-2427-2004, 2004.

Pratt, K., Custard, K., Shepson, P. *et al.* Photochemical production of molecular bromine in Arctic surface snowpacks. *Nature Geosci* 6, 351–356 (2013). https://doi.org/10.1038/ngeo1779

The detection limits of the ozone measurements, as well as the sensitivity of the MAX-DOAS vertical profile measurements, should be briefly discussed in Sections 2.1 and 2.2, respectively.

Both Sections have been updated and now include the detection limits of the ozone sondes (2 ppb in the boundary layer), Zeppelin instrument (1 ppb), and a brief discussion about the MAX-DOAS vertical profile sensitivity.

Section 2.1: It is not entirely clear how the number of ODEs is determined, in particular for the in-situ instrument. My understanding of a single ODE is a continuous period in time during which the ozone VMR remains below a certain threshold value. Here, it is not clear whether the number ozone depletion events are counted, or rather the number of hours during which ozone VMR remains below the threshold. This needs to be clearly defined. I think it would be inappropriate to count each hour of low ozone as a single ODE.

Thank you for pointing this out. In this study, consecutive hours below the threshold are each marked and individually counted as ODE, which is not really consistent with the definition of ODE. However, to avoid introducing a new abbreviation and jumping back and

forth between the new abbreviation and ODE, it has been decided to still use ODE. The following sentences have been included in Section 2.1 to avoid confusion:

*We here use the name ODE although it is not quite correct in this context, since during a longer ODE, all consecutive hours below the threshold are individually marked with the abbreviation ODE, which is not consistent with the definition of ODE. However, in order to avoid introducing a new abbreviation, we kept the term ODE when referring to individual hours having ozone values below the threshold.*

P5, L122: "The sensitivity to the choice of the threshold value": Sensitivity of what?

This sentence has been extended: *The sensitivity of the meteorological conditions and BrO [...]*

Figure 1 is very hard to read. In the left panel, it is impossible to recognize individual non-ODE profiles due to the large number of overlapping profiles, and it is hard to see any patterns in the time series shown in the right panel since the x-axis covers a large time range of more than 10 years. It is therefore impossible to recognize any seasonality. I would therefore appreciate if some other way of presenting the data could be found. For example, the vertical profiles could be shown as box-whisker-plots, and the time series could be shown as a separate figure with a larger width. For the time series, it appears that only springtime values are shown. I suppose there are also measurements during the rest of the year, and it would be nice to show all data in order to give an idea about the complete seasonality of ozone.

Following the suggestion of the reviewer, we have updated Figure 1. The data measured on Zeppelin mountain during the rest of the year has been inserted as black dots and is shown with a larger width.

Regarding the ozone sonde data, it is correct that it is not possible to read the individual profiles labeled as no ODE. The focus of this plot is on the individual vertical profiles of the ODE sondes and to have all the no ODE sondes for comparison in the background.

In order to perform a box-whisker-plot for the ODE sondes and the no ODE sondes as suggested, each ozone sonde measurement would have to be interpolated to the values of a fixed altitude grid. The information on the individual ODE profiles which we consider to be essential would be lost.  We agree that the figure is busy but still think that it is the better way to provide the important information.

P8, L174: I think it is not appropriate to call the (310-500) nm channel "visible" since light below 380 nm is not visible.

We agree and changed the formulation to: [..] *near-ultraviolet and visible (310–500 nm)* [..]

P8, L198: Can you be more specific with the location of the WRF domains, e.g. by providing coordinates of the centres of the domains?

An additional Figure has been inserted, containing the location of both WRF domains.

The "composite analysis" method described in section 2.7 is difficult to understand and this section should be re-written (see also general comments). A very simple approach

(anomaly = deviation from averages of maps of meteorological, chemical and sea ice parameters for ODE and non-ODE conditions) is described in a very complicated way:

- It does not become clear that the approach is applied to the spatial distribution of the observables over the entire Arctic region. During my first read, I thought this would refer to local parameters at the measurement site.
  The following sentence has been expanded:
  *To investigate the anomalies of 1. meteorological conditions, 2. BrO, and 3. SIC in the Arctic region during ODEs in Ny-Ålesund, a composite analysis was conducted.*

- Related to the ozone soundings, how do you define a "data point" (L216)? Is this the O3 VMR at a certain altitude (meaning that one profile consists of many data points), or a single ozone profile?
  Regarding the ozone sondes, a data point is defined as a single ozone profile. The word 'data point' as been removed, as it is misleading and a reference to Section 2.1 has been included, where the separation into ODEs an noODEs is described.

- It is not clear how you apply the threshold value to the ozone vertical profiles. Do you consider a measurement as ODE if the O3 VMR is below the threshold at some altitude, or does it need to be depleted over a certain altitude range?
  Described in Section 2.1.: All ozone sondes that contain ozone values below 15 ppb at altitudes between 0 and 2 km are marked as ODE and displayed in red.

- The number of ODEs stated in section 2.7 does not agree with the numbers in Table 2
  Changed in text to 1237 ODE and 24409 no ODE hours in the Zeppelin data set.

- How do you define an "ODE day" (L216)? An ozone depletion at the time of the balloon sounding does not necessarily mean that ozone is depleted during the entire day.
  The word 'day' is misleading and has been removed, since an ODE 'day' only covers the time of the ozone sonde measurement.

- The calculation of the anomalies 24 h and 48 h before and after the time of O3 observation is explained in a quite cumbersome way and should be rewritten.
  This part has been shortened and rewritten.

P9, L227: It is not mentioned that Y* is also calculated for the sea ice coverage parameter. It is not clear what you mean with "To obtain Y*(bar)_ODE, all selected Y* were averaged". How is this selection performed?

Y* and further the anomaly has not been calculated for sea ice, since it is not expected (and results not shown here confirmed it) that there will be much change in SIC during this time period.

With 'selected' all points in time 24 or 48 hours before/after an ODE were meant. This section has been rephrased.

P10, L260: It is not clear to me how you it can be concluded that "already ozone poor air is transported to the measurement site" if there are indications that recycling of Br_x on blowing snow took place. I think the opposite is likely as well, namely that saline particles are transported to the measurement site and ozone destruction took place all the time along the trajectory, and probably still takes place in situ.

*This sentence has been adapted: These findings indicate, that Br is likely recycled on aerosol or blowing snow on its way to Ny-Ålesund and therefore ozone is continuously depleted along the trajectory to and in Ny-Ålesund.*

P11, L262: Here you discuss an increase of the SIC on ODE days. It is hard to imagine that sea ice cover changes that rapidly, since sea ice formation is a very slow process, while ODEs occur over time scales of only a few hours.

The discussion of SIC is based on a publication from Aue et al., 2022 where they found a change in SIC in the Arctic region due to cyclones. Since we see this low pressure anomalies during ODEs, we thought that this might be also the case for ODEs. Strong winds and a cold air outbreaks could lead to fast sea ice change or formation. But as already mentioned, these anomalies might be due to seasonal effects. Camera images from Zeppelin mountain during the time of the case study showed freshly formed sea ice in the Kingsbay after the cyclone passed Ny-Ålesund.

Aue, L., Vihma, T., Uotila, P., & Rinke, A. (2022). New insights into cyclone impacts on sea ice in the Atlantic sector of the Arctic Ocean in winter. Geophysical Research Letters, 49, e2022GL100051. https://doi.org/10.1029/2022GL100051

Section 3.2.1: I suggest to skip this section as already detailed in the general comments

See answer in the general comments.

Sections 3.2.2 and 3.2.3 discuss seasonal and inter-annual variations of the anomalies. These are not sensitivity analyses, as the title of Section 3.2 suggests.

The title of Section 3.2. has been changed to: *Sensitivity and temporal analysis*

P17, L374: Please quantify the detection limit of the ozone measurements.

That has been done in the updated Section 2.1 (see 3[rd] comment).

Section 3.3: Here vertical profiles of ozone from balloon soundings are compared to vertical profiles of BrO from MAX-DOAS. It is speculated that blowing snow plays a role in the release of reactive bromine. To further support this hypothesis, it would be important to also show and discuss vertical profiles of the aerosol extinction, which should be available from the MAX-DOAS measurements.

We thank the reviewer for this suggestion which helped to better understand the evolution of this event! We checked the aerosol profiles and found enhanced aerosols only on April, 1[st.]. April 2[nd] and 3[rd] did not show any signs of enhanced aerosols in the profiles.

After April 1[st], a new layer of ice seemed to form in Kingsbay when looking at the camera images from Zeppelin Mountain, which might be contributing to the ozone depletion on the second and third, instead of the blowing snow as initially assumed. Therefore, ozone depletion from local Br is likely to happen on April 2[nd] and 3[rd]. The paragraph has been rewritten accordingly.

Figure 8: BrO does not seem to be present over the entire altitude range where ozone depletion is observed. Can you elaborate on the reasons for this discrepancy?

The main reason lies in the details of the BrO retrieval, which has been modified as described below:

Figure 8 shows slightly different vertical profiles of BrO compared to the initially published manuscript as we realized that profiles with test retrieval settings have been shown rather than the commonly used settings. The new profiles are slightly lifted to higher altitudes compared to the results from the previous manuscript and a double peak appears for the first shown day. As the sensitivity for higher altitudes is limited, the exact altitude for the maximum concentration cannot be completely constrained within a MAX-DOAS profile retrieval (see also comment below). The double peak indicates that BrO can be found in higher concentrations for almost the entire altitude range where also lower ozone values can be seen. Note that it is not possible to retrieve box-like features or sharp edges from MAX-DOAS measurements with optimal estimation based inversion algorithms due to a priori smoothing effects.

The vertical sensitivity of MAX-DOAS profile retrievals is highest for lower altitudes and decreases strongly for altitudes larger than 2-3km. However, elevated trace gas layers can still be retrieved when this layer is the dominant trace gas concentration - no shielding effect of larger near surface concentrations are present.

**Technical Corrections**

**If no further comment has been written, it should be considered as 'done'.**

P2, R1-R6: Chemical formulas should not be in italic

P4, L103: On can either discuss a case study or observe a case, but observing a case study does not make much sense (this would at the very most be meta-science).

P4, L110: I suggest to rewrite this sentence as follows: "The vertical resolved ozone sonde profiles allow to study the altitude distribution of ODEs in the boundary layer"

P4, L113: Add "described below" to the end of the sentence since the threshold values are not defined yet.

P4, L116: Insert "does" before "not necessarily".

P5, L118: Replace "enables" with "provides".

not changed → two times provide then in two consecutive sentences

P5, L122: It should be stated that the threshold value applies to the ozone VMR.

P5, L125: "The background level of ozone in the boundary layer is normally around 40 ppb".

P5, L126: This sentence can be deleted since the application of the threshold value is already explained at the beginning of the paragraph.

P7, L136: Please explain the acronym/abbreviation "AWIPEV"

removed AWIPEV

P7, L140: "sun light" -> "sunlight"

P8, L194: Replace "have" with "achieve"

Section 2.7: "Time point" is not a correct English term, I suppose you mean point in time or time of measurement.

P9, L220: "where" -> "when"

P21, L439: "the same pattern" -> "similar patterns"

P21, L452: "extend" -> "extent"

**References**

Benavent, N., Mahajan, A. S., Li, Q., Cuevas, C. A., Schmale, J., Angot, H., Jokinen, T., Quéléver, L. L. J., Blechschmidt, A.-M., Zilker, B., Richter, A., Serna, J. A., Garcia-Nieto, D., Fernandez, R. P., Skov, H., Dumitrascu, A., Simões Pereira, P., Abrahamsson, K., Bucci, S., Duetsch, M., Stohl, A., Beck, I., Laurila, T., Blomquist, B., Howard, D., Archer, S. D., Bariteau, L., Helmig, D., Hueber, J., Jacobi, H.-W., Posman, K., Dada, L., Daellenbach, K. R., and Saiz-Lopez, A.: Substantial contribution of iodine to Arctic ozone destruction, Nature Geoscience, 15, 770–773, https://doi.org/10.1038/s41561-022-01018-w, 2022.

---

## Author Comment (AC2)

**Comments from anonymous Referee #2:**

We would like to thank the reviewer for his/her helpful comments. We hope that we could address all questions and unclear points satisfactorily.

Legend: Author comments in blue, Referee comments in black.

**Review egusphere-2023-522**

Investigation of meteorological conditions and BrO during Ozone Depletion Events in Ny-Ålesund between 2010 and 2021

**1 general comments**

*evaluating the overall quality of the discussion paper*

The paper investigates the importance of polar cyclones and associated meteorological condition on ozone depleting events (ODEs) observed in the proximity of Ny-Ålesund. The authors apply complementary datasets of both ozone (O3) and BrO (e.g. in situ observations and various remote sensing products) and combine these with Lagrangian (FLEXPART) and Eulerian (WRF) modeling techniques. Though, no Chemistry Transport Model (CTM) or process modeling were applied in this study. Overall, the paper is

- well-structured,
- comprehensible in its application of the composite data analysis.

The language is
- comprehensive
- and for most parts only minor corrections may apply.

Though this paper was, overall, a real pleasure to read, a major shortcoming emerges from the distinction between background (no ODE) state of the atmosphere and ODE state. The authors define no ODE, in principle, by $Y_{noODE} = Y_{tot} - Y_{gap\ O3\ data} - Y_{[O3]<15\ ppb}$, where Y in this case refers to one record in 1-hourly meteorological data (e.g. $Y_{tot}$ would correspond to 744 records in March). Applying this definition of the background state of the atmosphere most likely reduces the explanatory power of the anomaly analysis because ODE conditions are likely to be contained in the no ODE data records. This clearly emerges from the sensitivity analysis in Section 3.2.1 where a lower threshold (20 ppb) was applied. This weakened the found dipole structure. The anomaly analysis could potentially benefit from stronger constraints on the background state, however, the general outcome of the study will probably not change significantly. Hence, either the authors, unconstrained by computational and human resources, repeat the anomaly analysis with a more clear-cut background state or discuss the implications more thoroughly in Section 3.2.1.

**2 specific comments**

*individual scientific questions/issues*

- p4 l116: "[...] not necessarily capture all ODEs at sea level.": Could you possibly give an estimate based on your composite analysis? Or give a number of coincident events?

  Due to the lack of continuous measurements in Ny-Ålesund at sea level, it is not possible to give an estimation. However, we compared the Zeppelin data with the 14 ozone sondes that measured ODEs. We found three ODEs in the ozone sonde data, which were not marked as ODE in the Zeppelin data. Two of them showed ODEs in the Zeppelin data shortly before/after the sonde launch, the third showed no sign of ODEs in the Zeppelin data at all. One out of the three ozone sondes has values above 15 ppb at the altitude of the Zeppelin station, the other two were below the 15 ppb threshold.

  When the remaining 11 ozone sondes marked as ODE were launched, ODEs were also visible in the Zeppelin data. There was no ozone sonde which was marked as no ODE, when an ODE was measured on Zeppelin mountain at the same time.

- p5 l121: "[...] ODEs occur mainly in polar spring [...]": Is there evidence that they do occur in, e.g., fall?

  Neither dataset shows ODEs (ozone < 15 ppb) in the fall and I am not aware of any publication on this. However, ODEs still occur from time to time in June, which can also be seen in the Zeppelin dataset (e.g., June 2013 (103 ODE hours), June 2014 (43 ODE hours)).

  To avoid confusion, the word 'mainly' has been removed from the sentence.

- p5 Table 1: Would it make more sense to differentiate between ODEs and total number of ascents in the respective month?

  This table should highlight the number of ODE and no ODE cases for different months and years, which will later form the basis of the composite analysis. Accordingly, it is more important here to show the number of ODEs and no ODEs and not to emphasize the total number of measurements. Adding additional columns with the total number of measurements would make the table too cluttered.

- p6 Table 2 (and respective paragraphs in the text): How is "ODE" count exactly defined here? It seems as if you count consecutive hours of below threshold O3 as one event each. An ODE can, in fact, as your correctly wrote and showed (Fig 7), last for several days. Hence, your definition of ODE in this Table is not consistent with, e.g., Section 3.3.

  It is correct that consecutive hours below the threshold are each marked with the abbreviation ODE, which is not consistent with the definition of ODE. However, to avoid introducing a new abbreviation and jumping back and forth between the new abbreviation and ODE, it has been decided to still use ODE, but the following sentences have been included in Section 2.1 to avoid confusion:

  *We here use the name ODE although it is not quite correct in this context, since during a longer ODE, all consecutive hours below the threshold are individually marked with the abbreviation ODE, which is not consistent with the definition of ODE. However, in order to avoid introducing a new abbreviation, we kept the term ODE when referring to individual hours having ozone values below the threshold.*

In any case, no ODE and ODE [hours with O3 below threshold] do not add up to the total number of hours in each month, e.g., 744 h in March. It is not clear if this is caused by gaps in data. Could you clarify this?

These are gaps in the Zeppelin data. E.g., in March and April 2016, and there are several smaller data gaps (hours to days) spread over the years.

In order to make this clear, the following sentence has been added: *Note that Zeppelin ozone data has several data gaps that result in an incomplete number of hours of data per month.*

- p5 l122: How did you define the 15 ppb threshold? Is it possible to clearly separate populations in a histogram of O3 monitoring (bi-modal distribution)?

Discussed below (p13 Section 3.2.1).

- p5 l129: How like do you detect "fake" ODEs due to low hemispheric O3 background especially at higher thresholds?

We cannot eliminate the possibility that some of the ODEs are fraudulent. Nevertheless, we think it is not likely that they were created by any mechanism other than depletion by halogens around the 15 ppb and 20 ppb threshold.

- p7 l154–177: "From 2010 until 2017 [...]": Too specific and technical. You may summarize the technical information about the different satellite products in a table and keep only additional information in the text.

The entire paragraph has been removed and replaced by a table. Some information from the deleted part has been added to the paragraph above.

- p8 l195–196: "[...] two domains were used in a two-way nested run, i.e. the values of the coarse domain are overwritten by values of the higher resolution [...]": That's not entirely correct, as changes in the inner and outer domain influence field information of one another in a two-way coupled model system. Either explain the coupling in detail or drop the second half of the sentence starting with "i.e.".

The second half of the sentence has been removed.

- p9 l219-223; eq (1): Y ODEanom = Y ODE − Y noODE : This method is slightly problematic in the sense that you may have erroneously detected ODEs / noODEs in both data sets (even more so in the data sets with the lowered threshold of 20 ppb). These will blur the signal you are after. In this context, the definition of ODE that you apply to the ozone monitoring data is not sound (see also comment regarding Table 2). Consecutive hours of below threshold ozone concentrations are a necessity to identify an "event", hence the number of hours of ODE is misleading with respect to ODE statistics. The lag analysis that you apply in the following should be associated to the onset/end of an individual ODE. In summary, if you use times of "no ODE" you have to make sure that these really do not contain any ODE which I currently cannot see satisfied in your analysis. Another threshold, e.g. for normal ozone concentrations, could do the trick.

The issue of the naming 'ODE' has been discussed above in p6 Table 2.

Regarding the second issues arising from the no ODE dataset erroneously containing ODEs: we performed a second run on the Zeppelin dataset, using a 15 ppb threshold for ODE (as in the article) and a 40 ppb threshold for no ODE. Any measurements between 15 and 40 ppb are not incorporated. The results are shown in the figures below:

[Figure]

Compared to the results shown in the paper (for the 15 ppb threshold), we can see that the overall pattern stays the same, but there are enhanced anomalies in BrO, MSLP, PBLH and temperature. Setting a no ODE threshold removes weaker ODEs, resulting in increased anomalies, but the overall pattern remains. The effect is similar to the result when changing the ODE threshold is changed.

These findings have been included into Section 3.2.1.

- p10 Fig 2: Regarding the shown data, where does the "white" area south of Svalbard come from? Did you use some kind of "sea ice edge" filtering to exclude these data or are there no data due to retrieval constraints?

  The satellite data has been sea-ice flagged to analyse only sea ice covered areas (described in Section 2.3). Hence no satellite data south of Svalbard was included in the analysis.

- p11 l274: "long computing times": On which kind of system? Personal computer, HPC? If former is the case an application of other computational resources should have been considered for this analysis, perhaps?

  A personal computer has been used and we agree that computing times are not an issue when using larger computing infrastructure.

- p13 Section 3.2.1: How did you derive the different threshold values? Are they "random" choices or were the derived from a bi-modal analysis of ozone concentration distributions if that is even possible? As you show in this section, the choice of threshold is crucial for identifying a causal signal.

[Figure]

- As shown in the Histogram above, it is not possible to derive a clear ozone threshold from a bi-modal analysis. The thresholds are more of a 'random' choice, based on several applied threshold values from other studies (see Halfacre et al., 2014: Section 2.2 and supplementary). Another criterion was to get a sufficiently large number of ODEs in the ozone sonde data set.

  Halfacre, J. W., Knepp, T. N., Shepson, P. B., Thompson, C. R., Pratt, K. A., Li, B., Peterson, P. K., Walsh, S. J., Simpson, W. R., Matrai, P. A., Bottenheim, J. W., Netcheva, S., Perovich, D. K., and Richter, A.: Temporal and spatial characteristics of ozone depletion events from measurements in the Arctic, Atmos. Chem. Phys., 14, 4875–4894, https://doi.org/10.5194/acp-14-4875-2014, 2014.

  Have you analyzed coincident ODEs in both sounding and monitoring data with respect to the meteorological and BrO conditions? Or are there too few coincident ODEs?

  We have not analysed coincident ODEs in both sounding and monitoring data. However, as described in p4 l116 there are 11 out of 14 coincident ODEs, and only one ODE in sounding data does not show any signs of ODE in the Zeppelin data. Therefore, the BrO and meteorological conditions for the coincident ODEs are most likely very similar to the results shown in Figure 2 and 3 for the ozone sondes.

- p14 l309: "Overall, the anomalies are slightly less pronounced when using the 20 ppb threshold.": This doesn't really come as surprise. By lowering the threshold you'd allow for more "false positive" ODEs that can originate from both transport and mixing of air parcels with different trace gas concentrations, as well as the inclusion of subsiding ODEs where the actual cause is not present in your time lag analysis.

  It is not clear what is meant by 'false positive' ODEs. Even though the threshold is higher, ozone levels are still 50% below the normal background value of 40 ppb. Even if the ozone is decomposed elsewhere, the ozone level upon arrival in Ny-Ålesund must remain below a certain threshold to be considered as an ODE. Therefore, the conditions for ozone depletion must be similar or the same there as well.

It is correct that the lower threshold also includes more points in time of subsiding ODEs, leading to a slight bias. But as already mentioned above, 20 ppb is well below a normal background value, so it can be assume that the ODE conditions are still present, albeit in a subsiding form.

Even though it did not come as a surprise that the anomalies are slightly less pronounced, we have not been able to come up with a satisfactory explanation yet.

- p17 l361–370: "Several years [...] show strong anomalies [...] other years still show similar patterns [...]": Let's turn this around: If you'd find similar patterns in your meteorological fields but no ODE, wouldn't that mean that these meteorological conditions do not suffice as cause of ODEs?

This is a legitimate question, but it is beyond the scope of this paper to answer adequately. For that, we would need to establish a routine which makes it possible to identify ODE meteorological conditions for the Arctic region and compare it to the Zeppelin ozone data. If similar meteorological conditions are recognized more frequently in spring, which however do not lead to an ODE, the influence of meteorology on ODEs would have to be reviewed again. However, it would have to be taken into account that other factors (sea ice coverage, amount of saline aerosols, etc.) can also play a role in ODEs.

- p17 l385–387: "[...] the amount of BrO is not fully captured by the satellite observations." Could this be due to the algorithm used to separate tropospheric and stratospheric columns?

The stratospheric correction is probably an issue alongside the satellite's problem of detecting local phenomena due to lack of sensitivity.

Following sentence has been changed: T*his leads to the assumption that the amount of* BrO *is not fully captured by the satellite observations, due to lack of sensitivity in detecting local phenomena.*

- p21 l452: "Due to a decrease of sea ice extend less source area for BEE will be available." Without taking the processes associated with BEE and ODE into account, this remains highly speculative. Given blowing snow on first year sea ice and brine on (young) sea ice is among the major sources of Br in the polar spring boundary layer (see e.g. 10.5194/acp-12-6237-2012 for a review of processes), the extent of the sea ice is probably less important compared to the structure and dynamics of the sea ice, higher wind speeds, and a change in frequency in the occurrence of polar lows. Dynamics of sea ice formation have been notoriously hard to detect with passive sounding satellite-born instruments, but advances might have been made in recent years.

This sentence has been removed as it is too speculative.

**3 technical corrections**

*purely technical corrections*

**If no further comment has been written, it should be considered as 'done'.**

- p2 l24: "during sunlight": term?

Changed into '*when sunlight is present*'

- p2 R1–R6: Typesetting of chemical formulas: $Br_2 \rightarrow Br_2$ , asf.

- p6 Fig 1: (left) Maybe indicate altitude of Zeppelin observatory?

- p7 l154 ff: "From 2010 until 2017 [...]": Duplicate of paragraph "To analyse [...]" (p7 l146 ff). Please condense the two paragraphs.

  See above p7 l154–177

- p8 l183–185: "Additionally, to analyse [...]": This sentence might be grammatically incorrect. Maybe better: *Daily AMSR (...) sea ice concentration (SIC) observations on a 25 × 25 km 2 grid have been used to analyse the SIC [...].*

- p8 l184, l191, l190: Style! "was used" is used in each of these consecutive sentences. You may want to rephrase.

- p8 l196–198 and Fig 9: Definition of WRF domains: Which projections were used? Does the clipping shown in Fig 9 represent the boundaries of the outer domain? Would it be possible to indicate the location of the inner domain in the WRF related plots in Fig 9?

- p9 l213, l214, l225, and others: "time points" Incorrect term. Rephrase → *times*.

- p9 l215–216: "To separate the ozone data [...]": Repetition of Section 2.1. Remove or rephrase: *From the ozone data (see Section 2.1), we found 14 ODEs and 228 no ODEs in ozone sonde data records and [...].*

- p9 l219 ff: Typesetting of formulas. If not defined otherwise in the journal's style guide, non-indexing subscripts should not be set in italic font: $Y_{ODE} \rightarrow Y_{ODE}$ .

- p10 Fig 2: "BrO VCD anomalies for ODE and no ODE" Caption text confusing? I would assume, you should drop "for ODE and no ODE" here. Regarding the shown data, were does the "white" area south of Svalbard come from? Did you use some kind of "sea ice edge" filtering to exclude these data or are there no data?

  "for ODE and no ODE" has been removed in the caption of Figure 2 and 3.

  The second part of this comment has been answered above in p10 Fig 2.

- p10 l235: Typo: "Ny-Alesund" → Ny-Ålesund

- p10 l242: Typo: "the the" remove one "the"; Missing comma after adverb?: "Normally the Icelandic low [...]"

- p10 l243: "Due to the lower pressure [...]": Perhaps low pressure system? (But I'm not firm in weather synoptics.)

- p15 l314–345: "[...] a more pronounced lower pressure anomalies [...]": Mismatching singular article and plural noun.

- p17 l378: "As shown in the blue line": preposition?: *As shown by the blue line*

- p17 l379: "in the same altitude": preposition?: *at the same altitude*

- p21 l452: "extend" → *extent*

- p27: Empty → Fig C1 should have appeared here.

- p25: Fig A5 is not referenced in the text.